# Interpretable Causal Representation Learning for Biological Data in the Pathway Space

**Jesus de la Fuente**[1,2], **Robert Lehmann**[3], **Carlos Ruiz-Arenas**[1], **Jan Voges**[1]
**Irene Marin-Goni**[1,4], **Xabier Martinez-de-Morentin**[1], **David Gomez-Cabrero**[3],
**Idoia Ochoa**[2,4], **Jesper Tegner**[3], **Vincenzo Lagani**[3,5,*] and **Mikel Hernaez**[1,6,*]

[1] CIMA University of Navarra, CCUN, IdiSNA, Pamplona, Spain.
[2] TECNUN, University of Navarra, San Sebastián, Spain.
[3] Biological and Environmental Science and Engineering Division, King Abdullah
University of Science and Technology (KAUST), Thuwal, Saudi Arabia
[4] Dept. of Molecular Pharmacology and Experimental Therapeutics, Mayo Clinic, MN, USA
[5] Institute of Chemical Biology, Ilia State University, Tbilisi 0162, Georgia
[6] Center for Data Science (DATAI), University of Navarra, 31008, Pamplona, Spain.
[*] Corresponding authors.

## Abstract

Predicting the impact of genomic and drug perturbations in cellular function is crucial for understanding gene functions and drug effects, ultimately leading to improved therapies. To this end, Causal Representation Learning (CRL) constitutes one of the most promising approaches, as it aims to identify the latent factors that causally govern biological systems, thus facilitating the prediction of the effect of unseen perturbations. Yet, current CRL methods fail in reconciling their principled latent representations with known biological processes, leading to models that are not interpretable. To address this major issue, we present **SENA**-discrepancy-VAE, a model based on the recently proposed CRL method discrepancy-VAE, that produces representations where each latent factor can be interpreted as the (linear) combination of the activity of a (learned) set of biological processes. To this extent, we present an encoder, **SENA**-$\delta$, that efficiently compute and map biological processes' activity levels to the latent causal factors. We show that **SENA**-discrepancy-VAE achieves predictive performances on unseen combinations of interventions that are comparable with its original, non-interpretable counterpart, while inferring causal latent factors that are biologically meaningful.

## 1 Introduction

Causal Representation Learning (CRL) has raised in recent time as a promising approach for identifying the latent factors that *causally* govern the systems under study (Schölkopf et al., 2021; Ahuja et al., 2023). Among other disciplines, CRL have been recently applied on biological systems, providing precise testable predictions on causal factors associated with disease or treatment resistance (Zhang et al., 2024; Lopez et al., 2023). These methods usually operate on mixture of observational and interventional biological data, exploiting the distributional shift caused by the interventions with the goal of retrieving the causal latent factors and, possibly, how they mutually interact with each other. Concomitantly, Perturb-seq (Dixit et al., 2016) data have emerged as an ideal testbed for these type of analyses. This technology allows the gene expression profiling of single cells both in their unperturbed state and when one or more genes are made functionally inoperative (e.g., through CRISPR knock-outs (KO) (Gilbert et al., 2014)). While generating expression profiles for thousands of cells across a variety of experimental conditions is indeed advantageous in the CRL context, the high dimensionality of Perturb-seq data presents notable challenges for these models.

Deep learning approaches have allowed to predict transcriptional outcomes of novel (combinations of) perturbations in Perturb-seq data (Roohani et al., 2024; Cui et al., 2024; Gaudelet et al., 2024), or of known perturbations on novel cell types (Lotfollahi et al., 2019). However, to the best of our

knowledge, there are only two works so far applying CRL to Perturb-seq data (Lopez et al., 2023; Zhang et al., 2024), and we argue that these models severely lack interpretability, as the reconstructed latent factors cannot be directly reconciled with known biological processes, yielding talent factors that are difficult to interpret. Attempts made so far to boost interpretability in these models include computing associations between the reconstructed latent factors and the activity of known processes (Lopez et al., 2023), or arbitrarily selecting genes as representative of each latent causal factor (Zhang et al., 2024). As also suggested in a recent review (Tejada-Lapuerta et al., 2023), using biological processes as prior knowledge *during* the reconstruction process, rather than afterwards, may indeed boost the interpretability of the resulting models.

**Related work.** In the context of interpretable Representational Learning (RL), recent years have seen extensive applications of variational autoencoders (VAEs) to single-cell applications (Lopez et al., 2018). Some of these approaches enable interpretable latent factors either by enforcing gene-cell correspondence during training (Choi et al., 2023), by performing pathway enrichment analysis on linear gene embeddings (Zhao et al., 2021), or by modifying the VAE architecture to mirror user-provided gene-pathway maps (Seninge et al., 2021; Gut et al., 2021; Lotfollahi et al., 2023; Niyakan et al., 2024; Ruiz-Arenas et al., 2024a). Importantly, while these latter methods apply architectural changes that are similar in spirit to the ones we propose in this work, none of them present strong theoretical guarantees for the *causal* interpretation of their embeddings.

Therefore, in this paper we show how CRL algorithms can be extended in order to employ biological processes (BPs) as prior knowledge, improving the interpretability of the resulting latent factors. We base our results on the CRL framework first introduced by Ahuja et al. (2023) and then expanded by Zhang et al. (2024). In particular, we present **SENA**-discrepancy-VAE, a CRL model based on the recently proposed discrepancy-VAE. We show that **SENA**-discrepancy-VAE yields predictive performances comparable to the ones of its original counterpart on unseen combination of perturbations, while providing a mapping between latent factors and biological processes. To this end, we modify the discrepancy-VAE's encoder architecture (Zhang et al., 2024) and embed it with biological processes as prior knowledge. To our knowledge, this is the first effort to reconcile CRL with biological interpretability, achieving both principled identifiability and interpretability of causal latent factors in the biological pathway space.

## 2 PRELIMINARIES AND BACKGROUND

### 2.1 CASUAL REPRESENTATION LEARNING

In what follows, we use the notation of Zhang et al. (2024), where we further use upper-case to denote random variables, lower-case to denote (inferred/observed) realizations of the random variables, upper-case bold to denote matrices, and lower-case bold to denote vectors. Let's assume that samples $x \in \mathbb{R}^n$ are generated according to a process governed by a set of latent variables $U \in \mathbb{R}^d$, where $d << n$. These latent variables are not required to be independent from each other. Instead, each latent factor $U_i$ may be regulated by a subset of other latent factors, namely its parents $Pa(U_i)$, according to a structural mechanism $U_i \leftarrow s_i(Pa(U_i), Z_i)$, where $Z_i$ is an exogenous variable independent of $Pa(U_i)$ and $Z_j$, $j \neq i$. The latent factors, as well as their possible regulatory relationships, are unknown.

In absence of interventions, the latent factors are sampled from the distribution $\mathbb{P}_U$ and the measurements $x$ are derived through a decoder function $g$, i.e., $u \sim \mathbb{P}_U, x \leftarrow g(u)$. Interventions are assumed to affect directly the latent variables $U$, rather than the observable $x$, and they can be either *hard* or *soft* (Pearl, 2009). In brief, *hard* intervention forcefully set the value of $U_i$ to a specific level, effectively severing any association between $U_i$ and its parents, while *soft* interventions solely modify the causal mechanism $s_i$, altering the relationship between the variable and its regulators. Thus, under intervention $I$, $U$ is sampled from a new distribution $u^I \sim \mathbb{P}_{U^I}$, while the decoder function $g$ remains unchanged, i.e., $x^I \leftarrow g(u^I)$.

In this context, the main goal of CRL is to identify a decoder function $h$ and encoder function $f$ such that $h \circ f(x) = x$ and $f(x) = \tilde{u}$, where $\tilde{u}$ "reconstructs" $u$ as accurately as possible, while $h$ approximates $g$. Additionally, one may be interested in identifying the regulatory (causal) mechanisms $s_i$ among the $U_i$ components.

Ahuja et al. (2023) proved that $u$ can be retrieved up to an affine linear transformation, i.e., $\tilde{u} = \boldsymbol{A} \cdot u + \boldsymbol{c}$ (Theorem 4.4 in Ahuja et al. (2023)). This requires two pivotal assumptions: (i) each intervention must targets a single component of $U$, and (ii) the decoder function $h$ is a full rank polynomial. Notably, multiple interventions can target the same latent component $U_i$, and, most importantly, the encoder function $f$ is only required to be non-collapsing.

Zhang et al. (2024) further expand on this framework and reach the notable result that *U can be retrieved up to permutation and scaling*, i.e., $\tilde{u}_j = \mathbf{a}_i \cdot u_i + c_i$ (Theorem 2 in Zhang et al. (2024)). This result requires that the relationships $U_i \leftarrow s_i(Pa(U_i), Z_i)$ can be represented by a Directed Acyclic Graph (DAG) with specific characteristics, and holds for both *hard* and *soft* interventions. Importantly, the authors proposed a VAE-based architecture, the *discrepancy-VAE*, that implements their theoretical results within a deep-learning framework. Describing the details of the discrepancy-VAE architecture is out of the scope of this work, however we note here a few of its characteristics that are instrumental for the proposed **SENA**-discrepancy-VAE:

- The encoder $f$ is implemented as a two-layer multilayer perceptron (MLP).

- Once trained, the model provides two additional pieces of information: (i) a deep structural causal model $\left( \mathcal{A}, \{s_i\}_{i=1}^d \right)$ where the graph's adjacency $\mathcal{A}$ encodes the parent set of each latent factor, while the matrix $\{s_i\}_{i=1}^d$ encodes the causal mechanisms (strength of interactions) (Pawlowski et al., 2020); and (ii) a map between each intervention and its target in the latent space, together with an estimate of the effect that the *soft* intervention has on $s_i$.

- The variational nature of the discrepancy-VAE allows to predict the effect of unseen double perturbations, provided that each single perturbation is available during training.

- The model trains both encoder $f$ and decoder $h$ only on unperturbed cells, while the perturbed samples are solely used for deriving the effect of the perturbations on the deep structural causal model.

## 2.2 Biological processes or Pathways.

A biological process or pathway (BP) can be thought as the set of concerted biochemical reactions needed to perform a specific task within the cell (Kanehisa & Goto, 2000; Ashburner et al., 2000). In the context of this work, we loosely identify a BP as the genes contained within it, discarding information regarding other molecules or interactions. From this point of view, BPs can be simply thought as gene sets, where these gene sets can overlap or even contain one another.

## 2.3 CRL in the context of Perturb-seq experiments.

In a Perturb-seq experiment, measurements $x$ are single cell expression profiles, with each $x_i$ representing the expression of a single gene $i$[1]. Interventions are genetic perturbations in which one or multiple genes have their functionality inhibited (through, for example, a genetic knock-out, KO (Dixit et al., 2016)). In this sense, Perturb-seq perturbations represent *hard* interventions on genes: once knocked out, the level of functionality of the targeted genes does not depend anymore upon the other genes that usually regulate it. Two observations on genetic perturbations that are relevant for our main result:

- Each perturbation likely affects several BPs at once. BPs are highly interconnected and genes are usually involved in several BPs at once.

- Gene KOs (i.e., *hard* interventions) leads to *soft* interventions in BP activity. Biological systems are very resilient, partly due to high level of redundancy in their regulatory circuits (Reed et al., 2024). This means that following a gene KO, other genes may partly assume the role of the suppressed gene, ensuring that the BP activity does not reach a halt, even if it is somewhat impacted.

---

[1]Here we will assume that these values have been normalized and scaled to the point where they can be considered laying in $\mathbb{R}^n$.

## 3 BIOLOGICALLY-DRIVEN CAUSAL REPRESENTATION LEARNING

The CRL framework discussed in section 2.1 requires the decoder $h$ to be a polynomial function. In contrast, a much wider modeling flexibility is granted for the encoder $f$, which must simply be a non-collapsing function. Thus, *the encoder $f$ can be built so it incorporates biological processes as prior knowledge*.

To achieve this, we propose a two-layer, masked multilayer perceptron (MLP) encoder, which we termed the **SENA**-$\delta$ (**S**pars**E** **N**etwork **A**ctivity) encoder (Figure 1). Let $\{\mathrm{BP}_1, \ldots, \mathrm{BP}_K\}$ be the gene sets corresponding to $K$ BPs. Let $\alpha_k$ indicate the activity level of the $k$-th BP, summarizing to what extent genes within the corresponding BP are activated (i.e., undergoing transcription). Then, the first layer of the proposed encoder connects the gene expression values $x$ with BP activity levels $\boldsymbol{\alpha}$:

$$\boldsymbol{\alpha} = \sigma\left((W \odot M)^T \cdot x\right), \tag{1}$$

where $W \in \mathbb{R}^{n \times K}$ are the layer weights, $\sigma$ is the activation function, $\odot$ denotes element-wise multiplication, and $M$ is a mask matrix defined as:

$$M_{i,k} = \begin{cases} 1 & \text{if gene } i \in \mathrm{BP}_k, \\ \lambda & \text{otherwise.} \end{cases} \tag{2}$$

Each BP activity is thus defined as a linear combination of the expression values of its respective genes. Unfortunately, it is known that the knowledge of the specific genes involved in BPs is seldom complete (Kunes et al., 2024). Thus, the tunable hyper-parameter $\lambda$ allows genes outside of the defined gene sets to contribute to the BP activity if enough evidence of their involvement is present within the data. To this extent, $\lambda$ should be set to a value small enough to discourage irrelevant contributions. Henceforth, we refer to this layer as the **SENA** layer.

The second layer follows a VAE-type architecture where a fully connected linear layer with two heads ($\mu$ and $\sigma^2$) generates the exogenous variables $Z_j$ as:

$$z_j \sim \mathcal{N}(\mu_j, \sigma_j^2); \quad \text{where} \quad \mu_j = \boldsymbol{\alpha}^T \boldsymbol{\delta}_j^{(\mu)}, \quad \sigma_j^2 = \boldsymbol{\alpha}^T \boldsymbol{\delta}_j^{(\sigma)}. \tag{3}$$

Thus, the mean and standard deviation will be a linear combination of pathway activities $\boldsymbol{\alpha}$, weighted by the parameters $\boldsymbol{\delta}_j^{(\mu)}$, $\boldsymbol{\delta}_j^{(\sigma)}$, learned by the corresponding MLPs. Modeling each latent factor as a linear combination of BP activities, which we denote meta-pathway activities $Z_j$, allows us to seamlessly combine the biological observation that each intervention may affects multiple BPs, with the CRL assumption (which provides identifiability guarantees) that each intervention must target only one latent factor (Figure 1). Modeling each latent factor as a single BP would set these two principles at odds with each other. We also note that each of the two layers of the **SENA**-$\delta$ encoder could be modeled as a more generic, non-linear function, simply by adding intermediate layers. We opted for a simpler architecture in order to prioritize interpretability over representational capabilities.

Most importantly, the **SENA**-$\delta$ encoder can be seamlessly plugged in the discrepancy-VAE architecture by substituting the original, fully connected MLP encoder. This modification guides the discrepancy-VAE architecture towards a more interpretable subsets of the original solution space. We named this resulting model as the **SENA**-discrepancy-VAE (Figure 1).

Finally, we note that the **SENA**-$\delta$ encoder associates BPs activities to the $Z_i$s, which in turn act as as exogenous variables for the $U_i$s, as noted in Section 2.1. The $U_i$s are the actual causal latent factors involved in the causal graph, and the input for the polynomial decoder (see Fig. 1). To avoid confusion between these two sets of latent variables, we refer to the $Z_i$s as meta-pathway activities (as each $Z_i$ incorporates the activity of several pathways), and to the $U_i$s as causal pathway archetypes. Appendix I further elaborates on the relationship between $U_i$s and $Z_i$s through the causal graph $A$.

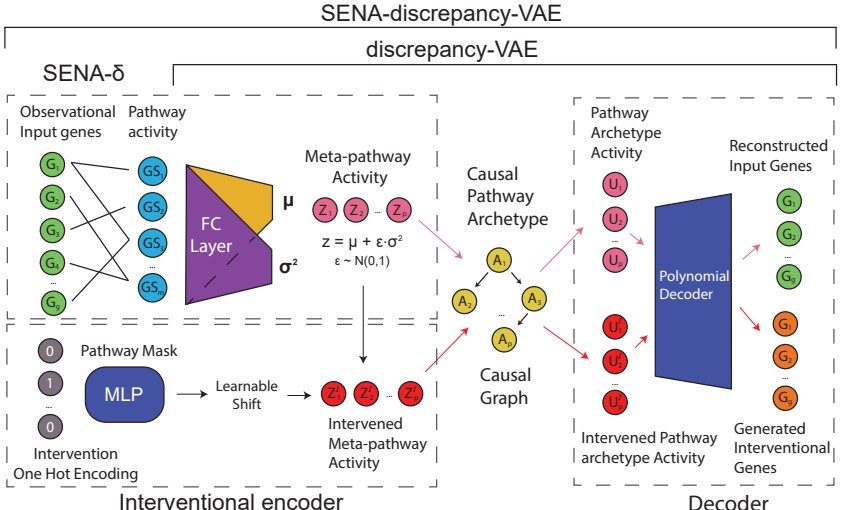

Figure 1: **Model overview. SENA**-discrepancy-VAE modifies the encoder of discrepancy-VAE to enforce a biologically-driven training through a pathway-based mask $M$.

## 4 EXPERIMENTAL SETTINGS

To first assess the learning capabilities of our proposed architecture, we performed several ablation studies using the proposed encoder within simple autoencoder (AE) and variational-AE (VAE) architectures. Our aim is to first assess whether the activity levels of the inferred (latent) BPs do indeed encode biological information (e.g., relevant activity changes are registered between perturbed and unperturbed data), and then to use these results to gauge the operational interval for $\lambda$. We then compare the **SENA**-discrepancy-VAE against the original discrepancy-VAE, over the task of predicting gene expression changes in novel combinations of perturbations, and finally analyze the interpretability of learnt latent causal factors.

### 4.1 DATA

We employ two large-scale Perturb-seq datasets, one collected on leukemia lymphoblast cells (K562 cell line) (Norman et al., 2019), termed the Norman2019 dataset, and a second one collected on acute myeloid leukemia cells (THP1 cell line)(Wessels et al., 2022), termed the Wessels2023 dataset. The authors of Norman2019 targeted 112 genes known to affect the growth of the K562 cells, yielding a total of 105 single-gene perturbation and 131 double-gene perturbations. The dataset underwent standard preprocessing steps for single cell data (filtering, normalization, and log-transformation (Wolf et al., 2018)), leading to a total of 8,907 unperturbed cells (controls), 57,831 cells under the 105 single-gene perturbations, and 41,759 cells under the 131 double-gene perturbations. The Wessels2023 study underwent the same preprocessing steps, ending up to include 424 unperturbed cells, 28 single-gene perturbations overall targeting 3036 cells, and 158 double-gene perturbations targeting 17592 cells. For all datasets we considered the 5,000 most variable genes in our analyses.

### 4.2 SELECTION OF BIOLOGICAL PROCESSES

Selecting an appropriate set of BPs is crucial, as the ideal selection should be sufficiently varied to include all active BPs in the system under study; and at the same time, it is desirable to reduce the redundancy that usually characterize large sets of BPs. Following Ruiz-Arenas et al. (2024b), we considered the Gene Ontology (GO) BPs (Ashburner et al., 2000), and selected GO BPs with less than 30 genes. We then discarded those with more than half of their genes in common with other selected BPs, as well as those with low replicability (Ruiz-Arenas et al., 2024b). We further refine this selection by including only GO BPs that have at least five genes represented in our input dataset, and by removing those that are ancestors of other terms within our list. This multi-step selection ensures that the final BPs are (mostly) non-overlapping and cover a large variety of biological processes.

### 4.3 IDENTIFYING ACTIVATED BIOLOGICAL PROCESSES

We exploit the architecture of the **SENA**-$\delta$ encoder to identify BPs that are activated under specific perturbations. In particular, we expect that the activated $\text{BP}_k$ should have its inferred activity level $\alpha_k$ significantly altered with respect to unperturbed controls for those perturbations targeting genes within that BP. To measure this effect, we define the differential activation (DA) for $\text{BP}_k$ under intervention $p$ as $\text{DA}_k^p = |\bar{\alpha}_k^p - \bar{\alpha}_k^c|$, where $\bar{\alpha}_k^p$ and $\bar{\alpha}_k^c$ are the values of activation function for $\text{BP}_k$ averaged over all perturbed and control cells, respectively. Because it is a difference among mean values, a t-test or similar inferential statistic can be used for assessing its statistical significance.

We then built two metrics for assessing to which extent the differentially activated BPs (i.e., statistically significant $\text{DA}_k^p$ values) are biologically meaningful. First, for each intervention $p$ we define $\mathcal{W}_p$ as the set of BPs that contain the targeted gene $i$: $\mathcal{W}_p = \{\text{BP}_k | M_{i,k} = 1\}$. The remaining (not affected) BPs are indicated as $\overline{\mathcal{W}}_p$. Intuitively, we would expect BPs containing the targeted gene to be the most affected by the intervention, while the other processes should only suffer indirect effects.

We then define the metric Hits@N, as the percentage of BPs in $\mathcal{W}_p$ that are ranked within the first N positions in terms of $\text{DA}_k^p$. The parameter N is set to 100 in our analyses. Let $R_k^p$ be the rank for $\text{BP}_k$ under perturbation $p$ according to $\text{DA}_k^p$. Then, Hits@N is defined as:

$$H_N^p = \frac{1}{|\mathcal{W}_p|} \sum_{k \in \mathcal{W}_p} \mathbb{I}[R_k^p \leq N]. \tag{4}$$

Finally, we define the differential activation ratio (DAR) for a perturbation $p$ as:

$$\text{DAR}_p = \frac{|\overline{\mathcal{W}}_p| \sum_{k \in \mathcal{W}_p} \text{DA}_k^p}{|\mathcal{W}_p| \sum_{k \in \overline{\mathcal{W}}_p} \text{DA}_k^p}. \tag{5}$$

This ratio contrasts the average activation for BPs directly affected by the perturbation against the average of the remaining BPs. Although computing this metric involves aggregating pathways with varying numbers of targeted genes, the imposed minimum of five genes per gene set and the definition of DA as an activation ratio make this metric potentially robust to intrinsic noise within the pathways. Note that both metrics require $\mathcal{W}_p$ and $\overline{\mathcal{W}}_p$ to contain at least one BP.

## 5 ABLATION STUDY

Due to the high sparsity infused in the **SENA**-$\delta$ encoder, one cannot assume that such encoder has good reconstruction capabilities while maintaining interpretability. Moreover, we also seek to understand the reconstruction-interpretability trade-off driven by the $\lambda$ parameter. Hence, in what follows we assess the **SENA**-$\delta$ encoder by employing it in an AE and VAE architectures (with MLP as decoders in both cases), and compare it with a fully-connected encoder (denoted MLP) and two $\ell_1$-regularized encoders with $\lambda$ as the regularization parameter. To perform a fair evaluation, these architectures will each present two fully connected layers at the encoder, and the $\ell_1$ encoders will only have the first layer regularized to imitate **SENA**'s sparsity. To this end, we used the Norman2019 dataset. We evaluated the aforementioned architectures for several values of $\lambda$: $\{0, 0.1, 0.01, 10^{-3}\}$. Overall, four different aspects were assessed: data reconstruction and generative capabilities (for VAEs), and interpretability and sparsity of latent dimensions, described next.

**Data reconstruction and generative capabilities.** We evaluated the reconstruction and generative capabilities of the proposed models by computing the test Mean Squared Error (MSE) and Kullback–Leibler divergence ($\text{D}_{\text{KL}}$), respectively. The latter is only used in the variational architectures.

**Interpretability and sparsity of latent dimensions.** We evaluated the interpretability by measuring how differentially activated (DA, see above) the affected neurons (i.e. BPs containing the knock-out gene) are when compared to the rest of the neurons after the **SENA** layer. Hence, we compute the Hits@100 metric to measure the percentage of affected neurons in the top 100 DA BPs. Moreover, we define the sparsity of a model as the percentage of neurons with activation value (i.e., $|\bar{x}_i \cdot W_{i,k}|$) smaller than $10^{-8}$, which measures the contribution of every input gene to each factor after the **SENA** layer, and $\bar{x}$ refers to the mean expression across samples (cells). Reported metrics were computed on test samples.

**Results.** Overall, enabling residual connections between genes and BPs in a fully-connected fashion ($\lambda = \{10^{-2}, 10^{-3}\}$) maintains biologically-meaningful latent factors (Appendix IV Fig. 8-A) while yielding reconstruction capabilities in par with the fully-connected MLP (Appendix IV Table 4 & Fig. 9-B). The reason could be that these models present an efficient use of the model weights (Appendix IV Fig. 8-B), underscoring the relevancy of gene-BP relationships in Perturb-seq data. Interestingly, higher values of lambda ($\lambda = 0.1$) presented better reconstruction capabilities than the MLP (Appendix VII Table 4), at slightly sparser encoder (and hence, more interpretable) than the MLP (Appendix VII Fig. 8-B). On the other side, and as expected, $\lambda = 0$ presented highly interpretable latent factors at the cost of a significant drop in reconstruction capabilities. Additionally, our results show that the $\ell_1$-regularized MLPs does not perform well nor do they provide interpretable latent factors. Of note that when the analysis was perform only using the **SENA** layer, similar insights were obtained (Appendix VII Table 3 & Fig. 9-A). Finally, regarding the generative capabilities assessed on VAEs, the models based on **SENA**-$\delta$ encoder clearly outperforms other encoders on $D_{KL}$, with lower values of $\lambda$ being the best performing ones (Appendix VII Tables 4 and 3, VAE-based column). These results were generated enforcing that every BP contains at least 5 genes. Different such thresholds modifies the number of considered BPs in the **SENA** layer (Appendix IV Fig. 10-B), slightly affecting the learning capabilities (Fig. 10-A) and performance time (Fig. 10-C).

## 6 LEARNING INTERPRETABLE LATENT CAUSAL FACTORS

We next benchmarked **SENA**-discrepancy-VAE against its original counterpart, to assess the modeling and predictive capabilities of both models. This section focuses on the results obtained on the Norman2019 dataset, while Appendix V reports the results on the Wessel2023 data. For both datasets, we trained both models on the unperturbed and single-gene perturbations samples from Norman et al. (2019) (the latter are only used as a ground-truth for the MMD loss). We also benchmarked GEARS (Roohani et al., 2024), a state-of-the-art approach for multigene perturbation prediction. Double-gene perturbations were set aside for evaluation purposes. We train both models across 3 different runs with default settings (Appendix F of Zhang et al. (2024)). Given the good results (in interpretability and reconstruction performance) obtained in the ablation study (Section 5), we varied the number of latent factors within $\{5, 10, 35, 70, 105\}$, and the $\lambda$ for the **SENA**-discrepancy-VAE in $\{0, 0.1\}$ (Appendix VII Fig. 12 shows gradients and mask ($M$) distribution across several $\lambda$ values).

### 6.1 PERFORMANCE BENCHMARKING

Table 1 reports the results of the comparison, where MMD (Max Mean Discrepancy (Gretton et al., 2012)) measures the difference between the generated and true double-perturbation distributions. We report the average MMD over all 131 double-gene perturbations. Additionally, MSE indicates the reconstruction error for control samples during training, $D_{KL}$ is the variational loss (Kingma & Welling, 2014), and L1 := $||\mathcal{A}||_1$ represents the sparsity of the deep structural causal model.

Table 1: Benchmarking **SENA**-discrepancy-VAE and discrepancy-VAE on double perturbations prediction. Values are reported as mean ± variance computed on 5 runs with different initializations.

| Encoder | Metric | Latent Dimension | | | | |
|---|---|---|---|---|---|---|
| | | 105 | 70 | 35 | 10 | 5 |
| Original MLP | MMD↓ | 1.59811 ± 0.012110 | **1.73486** ± 0.012115 | 1.98993 ± 0.013053 | **2.43440** ± 0.030570 | **2.53237** ± 0.017662 |
| | MSE↓ | 0.02152 ± 0.000156 | **0.02298** ± 0.000064 | 0.02499 ± 0.000008 | 0.02699 ± 0.000079 | **0.02792** ± 0.000018 |
| | KLD↓ | 0.00022 ± 0.000008 | 0.00021 ± 0.000005 | 0.00021 ± 0.000003 | 0.00024 ± 0.000023 | 0.00030 ± 0.000027 |
| | L1↓ | 0.06097 ± 0.003718 | 0.06934 ± 0.002055 | **0.06714** ± 0.003649 | **0.07201** ± 0.009314 | 0.08319 ± 0.010590 |
| SENA-$\delta_{\lambda=0.1}$ | MMD↓ | **1.58489** ± 0.010610 | 1.75332 ± 0.004884 | **1.94984** ± 0.027883 | 2.49881 ± 0.056160 | 2.60439 ± 0.100708 |
| | MSE↓ | **0.02134** ± 0.000047 | 0.02300 ± 0.000120 | **0.02462** ± 0.000041 | **0.02688** ± 0.000076 | 0.02805 ± 0.000135 |
| | KLD↓ | 0.00019 ± 0.000001 | 0.00019 ± 0.000008 | **0.00019** ± 0.000003 | **0.00020** ± 0.000007 | **0.00020** ± 0.000002 |
| | L1↓ | **0.05243** ± 0.001541 | **0.05323** ± 0.003540 | 0.07308 ± 0.002529 | 0.08589 ± 0.010121 | **0.06856** ± 0.013631 |
| SENA-$\delta_{\lambda=0}$ | MMD↓ | 1.74588 ± 0.003579 | 1.90425 ± 0.014918 | 2.22812 ± 0.015438 | 2.62393 ± 0.056431 | 2.90044 ± 0.174605 |
| | MSE↓ | 0.02312 ± 0.000048 | 0.02460 ± 0.000078 | 0.02634 ± 0.000036 | 0.02800 ± 0.000236 | 0.02879 ± 0.000147 |
| | KLD↓ | **0.00018** ± 0.000002 | **0.00018** ± 0.000000 | **0.00019** ± 0.000002 | 0.00021 ± 0.000017 | 0.00022 ± 0.000021 |
| | L1↓ | 0.05418 ± 0.001903 | 0.05749 ± 0.000487 | 0.06730 ± 0.003195 | 0.08651 ± 0.025800 | 0.09486 ± 0.038286 |
| GEARS | MMD↓ | 14.9420 ± 0.233957 | 12.6036 ± 0.745302 | 13.2590 ± 0.559124 | 12.9774 ± 0.751373 | 15.3099 ± 0.191833 |

Interestingly, and despite the restrictions imposed by the **SENA**-$\delta$ encoder that could potentially decrease the **SENA**-discrepancy-VAE representational capabilities, the proposed model outperformed the MLP encoder for some latent dimensions in terms of MSE and MMD computed on unseen double perturbations for small values of $\lambda$ (0.1). Moreover, setting $\lambda = 0$ allowed the **SENA**-discrepancy-VAE to surpass the original MLP encoder on the $D_{KL}$ metric, while the optimal model for causal graph sparsity (L1) varied with latent dimensions. These results, which align with those of the ablation studies, highlight the potential of **SENA**-discrepancy-VAE. On the other hand, GEARS failed to properly model the evaluated double perturbations. Note however that GEARS does not provide a causal graph nor is a generative model (details in Appendix VI).

## 6.2 VISUALIZING **SENA**-DISCREPANCY-VAE LATENT FACTORS

We first investigated the association between perturbations and latent factor activation (Fig. 14). Both models tend to activate few latent factors. Specifically, the discrepancy-VAE model activate 8 to 9 factors across all perturbations when 35 or more latent factors are included in the model. These numbers decrease to 6 and 4 when 10 and 5 latent factors are available, respectively. At the same time, more than half of the perturbations are assigned to only 1 or 2 latent factors, creating a quite unbalanced mapping. The **SENA**-discrepancy-VAE follows a similar pattern. This seems to indicate that relatively few latent factors are needed for capturing the changes induced by perturbations, while the remaining latent factors assist in representing the overall distribution of gene expression data.

**Interpretation of the SENA-discrepancy-VAE latent factors.** The proposed model offers the possibility of inspecting its encoder for deriving the BPs composing the latent factors. By construction, each perturbation will target a single latent factor $U_i$, which enable us to associate each BP to the intervention with the largest differential activation value. Only significant differential activation values are taken into account (ranked within the top $1\%$ in absolute value, and a false discovery rate (FDR) $\leq 0.05$ via two-tailed t-test with BH correction). Figure 2 represents the causal graph for the latent factors associated to at least one BP for the **SENA**-discrepancy-VAE model with 105 latent dimensions and $\lambda = 0$. Ten edges with the highest coefficients in absolute value are reported for readability. Latent factors are represented as a word cloud of BPs (i.e., a graphical representation of the terms fre-

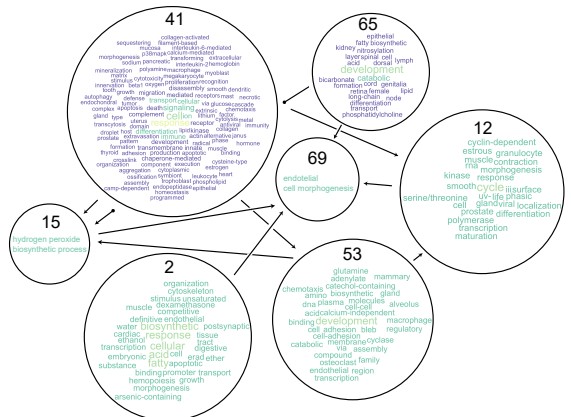

Figure 2: **SENA**-discrepancy-VAE causal graph on the Norman2019 data. Latent factors are represented by word clouds of associated BPs. Arrows indicate causal influences.

quencies within the BPs names). Table 7 (Appendix VII) reports the number of perturbations and BPs assigned to each factor and Table 9-10 (Appendix VII) shows the mapping between BPs and selected latent factors. It is worth noting that the inferred causal graph is robust across latent dimensions and $\lambda$s, where most of the inferred connection are maintained (Appendix II).

Latent factor 15 is targeted by perturbations on the JUN gene, and is associated with the activity level of GO:0050665, "hydrogen peroxide biosynthetic process". While JUNE is not included in GO:0050665, this gene is known to react with over-expression to oxidative stress (Vandenbroucke et al., 2008). For latent factor 69, the targeted PTPN13 gene is known to be involved in several tumors (Mcheik et al., 2020), thus it is not surprising to found it associated with a BP related to blood vessel formation. Interpretation of other factors requires careful inspection of their associated BPs (see Appendix VII Tables 9 & 10). For example, most of the BPs in latent factor 65 are associated to tissue development, while latent factor 12 contains several BPs related to protein activity.

Upon inspecting the connections on the causal graph, a first important connection is the one between factor 15, "hydrogen peroxide biosynthetic process"(Appendix VII, Table 9, third latent factor) which causes factor 69, "endothelial cell morphogenesis" (Appendix VII, Table 10, last row). It is well known that hydrogen peroxide stimulates endothelial cell proliferation (Stone & Collins,

2002; Anasooya Shaji et al., 2019). Thus, our causal graph captured this regulatory relationship in a fully unsupervised, data-driven way. In turn, factor 53 causally influences factor 15, and factor 53 contains the biological process "catechol-containing compound biosynthetic process" (Appendix VII, Table 10, second element of factor 53). It is well known that H2O2 can be produced by the metabolism of catecholamines (Noble et al., 1994; Seregi et al., 1982). An even more direct connection exists between latent factor 69 and latent factor 2, with the latter including "negative regulation of endothelial cell apoptotic process" (Appendix VII, Table 9, second row) among its biological processes. Taken together, these findings provide evidences for the correctness of our approach and its capability of recapitulating known biological causal relationships.

### 6.3 SENA-DISCREPANCY-VAE CAPTURES BIOLOGICALLY MEANINGFUL PATTERNS

We next evaluate the ability of the proposed encoder to maintain biologically-driven factors (see Appendix I). For this evaluation, we focus on the Norman2019 dataset, and we set the latent space dimension to 105, one for each single-gene perturbation in the dataset. We then evaluated the 37 knocked out genes that were present at the input. For each of these, we computed the DA score across all BPs (after the SENA layer) and found that those including the targeted gene reported higher DA on average (Appendix VII Fig. 13). Since each perturbation presented a different number of affected BPs, we next focused on the perturbations with the largest amount of targeted BPs (i.e., 7), and evaluated the significance (*statsannotation* package (Charlier et al., 2022)) of the DA among affected and not affected BPs (Mann-Whitney U test with BH p-value correction). Fig. 3-A shows the aforementioned analysis for the knock out genes LHX1, SPI1 and TBX3. This analysis highlighted the perturbation samples from LHX1 and TXB3 KOs as those presenting highly differentially activated BPs, validating the capacity of the proposed encoder to identify biologically-meaningful factors.

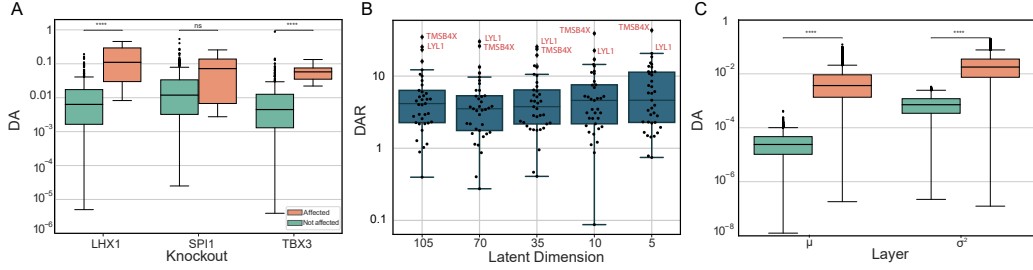

Figure 3: **SENA, $\mu$ and $\sigma^2$ layers interpretability analysis**. **A**. DA score for the three perturbations presenting the highest number of affected BPs among $\mathcal{W}_a$ and $\mathcal{W}_{\bar{a}}$. **B**. DAR of the 37 analyzed perturbations when varying the latent space dimensions. For every dimension, genes with the highest DAR are highlighted. **C**. DA score across the evaluated perturbations at the output of the $\mu$ and $\sigma^2$ layers among affected and not affected BPs. **ns**: non-significant, **\*\*\*\***: p-value $< 10^{-4}$.

To analyze the robustness of the **SENA**-discrepancy-VAE's encoder, we repeated the above analysis across several latent space dimensions, computing the DAR (Eq. 5) between those BPs containing the targeted gene and the rest. Once again, we found that almost every evaluated knock out gene reported a DAR > 1 along evaluated latent dimensions (Fig. 3-B). It is worth highlighting TMSB4X and LYL1, which reported a DAR $\gg 10$ consistently, indicating that the mean difference in activation of the **SENA**-layer neurons among perturbation and control samples for the BPs containing targeted genes was $\gg 10x$ times greater than for the rest. This underscores the capacity of **SENA**-discrepancy-VAE to drive the training process while maintaining biologically-meaningful factors.

**Interpretable $\mu$ and $\sigma^2$ layers.** We next assessed how the above shown interpretability is propagated through the **SENA**-discrepancy-VAE encoder. To this end, we performed the DA analysis on the output of **SENA**-$\delta$ encoder (Figure 1), where we measured the contribution of affected and not affected BPs from the previous layer to every neuron $j$ in the $\mu$ and $\sigma^2$ layers. To this end, we define the DA score for perturbation $p$ and BP $k$ at the $j$-th neuron of $\mu$ and $\sigma^2$ layers as $(\mathrm{DA}_k^p)_j = |\delta_{kj}| \cdot |\bar{\alpha}_k^p - \bar{\alpha}_k^c|$, where superscripts $p$ and $c$ denote perturbed and controlled activities, respectively.

Fig. 3-C shows the DA score on $\mu$ and $\sigma^2$ layers for affected and not affected BPs, following the same significance tests performed above. Again, this highlights that the **SENA**-discrepancy-VAE encoder maintains biologically-meaningful the **SENA**-$\delta$ encoder (Fig. 1), yielding interpretable exogenous variables $Z_i$, which we denoted the meta-pathway activities and also contributes to this differential activation. We next analyzed the biological significance of the meta-pathway activities, since these connections are learned in a data-driven manner. For this, we measured, through a permutation test, the contribution of the level 2 GO pathways (i.e., the parental GO terms of the used BPs in the **SENA** layer) to each of meta-pathway activity. Interestingly, multiple meta pathways were significantly associated to few level 2 pathways, underscoring our model capabilities to learn biologically-meaningful patterns at both high (BPs) and broad (meta-pathway) granularities (Appendix III.)

Finally, we performed a differential activation score analysis on the Norman2019 dataset after training **SENA**-discrepancy-VAE. We selected the top 6 largest DA scores, which belonged to 5 unique KO genes and gene sets, respectively (Table 8). Fig. 4-A shows the UMAP components of all intervened cells (in the input gene space) across the aforementioned genes, while Fig. 4 B-F depicts those cells colored by the DA score that each cell has on the evaluated gene set (GO term). Surprisingly, from the top 6 DA scores, we found that only GO:0038065 is initially targeted by COL1A1, while the remaining gene sets are

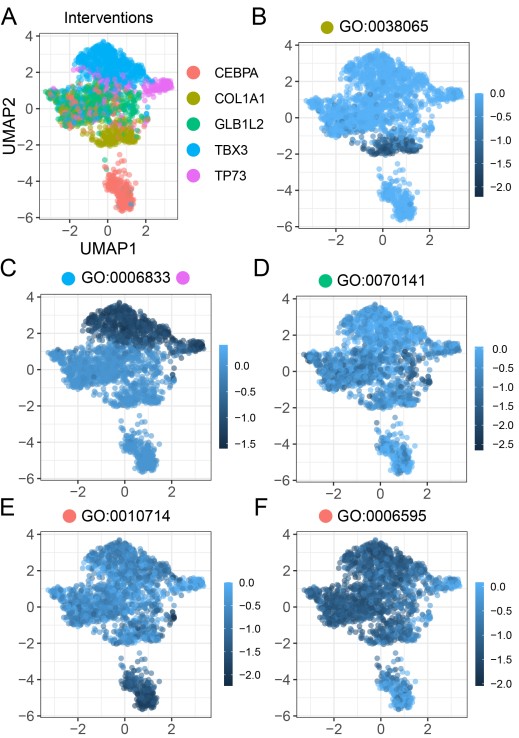

Figure 4: **DA score of 6 most significant (KO, BP) pairs**. **A**. UMAP of all intervened cells across the KO genes presenting the 6 largest DA scores. **B-F**. UMAP from **A** colored by the differential activation score on each cell across genes within each evaluated BP.

reporting specific-highlight on the cells belonging to the intervened KO without being directly targeted. For instance, the gene set GO:0006833 (Fig. 4-C) is activated by the TBX3 gene (same with GO:0010714 and CEBPA) without being explicitly encoded in the **SENA**-$\delta$ encoder. These underscore the potential of **SENA**-discrepancy-VAE to naturally learning biologically-driven patterns without specifically enforcing them.

## 7 DISCUSSION AND CONCLUSIONS

In this work we have demonstrated how biological processes can be used as prior knowledge in the context of causal representation learning. The resulting model, **SENA**-discrepancy-VAE[2], is on par, or even outperforming it in specific scenarios, in terms of predictive capabilities with the original discrepancy-VAE, while at the same time producing embeddings that can be easily inspected for assessing their biological meaning.

Among the the several findings reported in this study, it is striking that both models tend to assign most interventions to a small number of latent factors (see Fig. 14 for perturbation-to-latent factor associations in the Norman2019 data). Reasoning in terms of biological processes helps understanding why. The theory behind discrepancy-VAE requires that each intervention must be assigned to a single factor. Thus, this factor must represent all BPs affected by that intervention. If two (or more) interventions affect overlapping sets of BPs, then by necessity all these BPs must be mapped to the same factor. Overall, it may be argued that assuming that each intervention targets a single latent factor does not allow CRL methods to thoroughly disentangle the interplay between BPs, perturbations and latent factors. Thus, future CRL works should attempt to overcome this assumption.

---

[2]Python package, including data and code for reproducibility: github.com/ML4BM-Lab/SENA

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

# APPENDIX

## I INTERPRETABILITY OF LATENT FACTORS AND CAUSAL GRAPH THROUGH OUR PROPOSED SPARSE LAYER

In the variational autoencoder proposed at Zhang et al. (2024), the exogenous variable $Z_j$ is sampled from a normal distribution, where the mean and standard deviation of this distribution is defined by the fully connected layers in their encoder. In the proposed **SENA**-discrepancy-VAE (Fig. 1), the mean and standard deviation will be a linear combination of pathway activities $\boldsymbol{\alpha}$, weighted by the parameters $\boldsymbol{\delta}_j^{(\mu)}, \boldsymbol{\delta}_j^{(\sigma)}$, learned by the corresponding MLPs. Thus,

$$\mu_j = \boldsymbol{\alpha}^T \boldsymbol{\delta}_j^{(\mu)}, \quad \sigma_j^2 = \boldsymbol{\alpha}^T \boldsymbol{\delta}_j^{(\sigma)} \tag{6}$$

which then define the meta-pathway activities $z_j$ as $z_j \sim \mathcal{N}(\mu_j, \sigma_j^2)$. This allows the expectation of $Z_j$ to be interpretable, as

$$\mathbb{E}(Z_j) = \mu_j = \boldsymbol{\alpha}^T \boldsymbol{\delta}_j^{(\mu)} \tag{7}$$

However, we would also like this interpretablity to hold when going from the exogenous variables (the meta-pathway activities) to the causal factors $U$ (causal pathway archetypes). The latter are defined as $U = Z^T \cdot (I - A)^{-1}$, where

$$L \triangleq (I - A)^{-1} = \sum_{l=0}^{\infty} A^l = (I + A + A^2 + \cdots + A^K), \tag{8}$$

according to the Neumann series, and given that A represents the adjacency matrix of a Direct Acyclic Graph, with $L$ being the largest path (hence, $A^k = 0$, $k = K + 1, \ldots$). Here $A$ defines the causal relationships in the latent space. Therefore, the $j$-th causal factor can be expressed as

$$U_j = Z^T \cdot L_j,$$

where $L_j$ is the $j$th column of $L$, and encodes the number of path of at most length $K$ that ends at node $j$ in the causal graph. Hence, the expectation of the causal factor $U_j$ is given by

$$\begin{aligned} \mathbb{E}(U_j) &= \mathbb{E}(Z^T \cdot L_j) \\ &= \mathbb{E}(Z^T) \cdot L_j \\ &= \boldsymbol{\mu} \cdot L_j \tag{9} \\ &= \boldsymbol{\alpha}^T \cdot \boldsymbol{\Delta}^{(\mu)} \cdot L_j \tag{10} \\ &= \boldsymbol{\alpha}^T \cdot \tilde{\boldsymbol{\delta}}_j^{(\mu)}, \end{aligned}$$

where Eq. (9) and Eq. (10) from Eq. (7), and $\boldsymbol{\Delta}^{(\mu)}$ is the (learned) linear mapping between the pathway activities $\boldsymbol{\alpha}$ and the meta-pathway activities $Z$. Thus, $\tilde{\boldsymbol{\delta}}_j^{(\mu)}$ (linearly) maps the causal latent factors with the pathway activity scores through the learnt causal structure $L_j$, providing the mechanism for the interpretation of the latent factors, termed in our work as the pathway archetype activities.

Finally, we experimentally validated, using the Norman2019 dataset, that Eq. (9) holds for both the original discrepancy-VAE (MLP), and the proposed **SENA**-$\delta$ model for both $\lambda = \{0, 0.1\}$. We used all available unperturbed cells (ctrl) and 9 randomly-chosen perturbed cells, setting the latent dimension to the number of available perturbations in the dataset, i.e. 105. To this end, we first computed $\mathbb{E}(U_j)$ for every latent dimension $j \in \{1, \ldots, 105\}$ by forwarding the cells and averaging (over 10,000 realization of $Z \sim \mathcal{N}(\boldsymbol{\mu}, \boldsymbol{\sigma}^2)$) the obtained pathway archetype scores (i.e., the causal latent factors $U$s). On the other hand, we multiplied the BP activity scores ($\boldsymbol{\alpha}$) with the BP-to-meta-pathway mapper ($\boldsymbol{\Delta}$) and the causal mechanism ($\boldsymbol{L}$, Eq. 8). Both computations should be equal according to Eq. (10). Fig. 5 depicts for every type of perturbation and model (MLP, **SENA**-$\delta_{\lambda=0}$ and **SENA**-$\delta_{\lambda=0.1}$) both terms of the equality, as well as the Pearson's correlation. There is a perfect correlation among these two terms, and this patterns is maintained across models and perturbations.

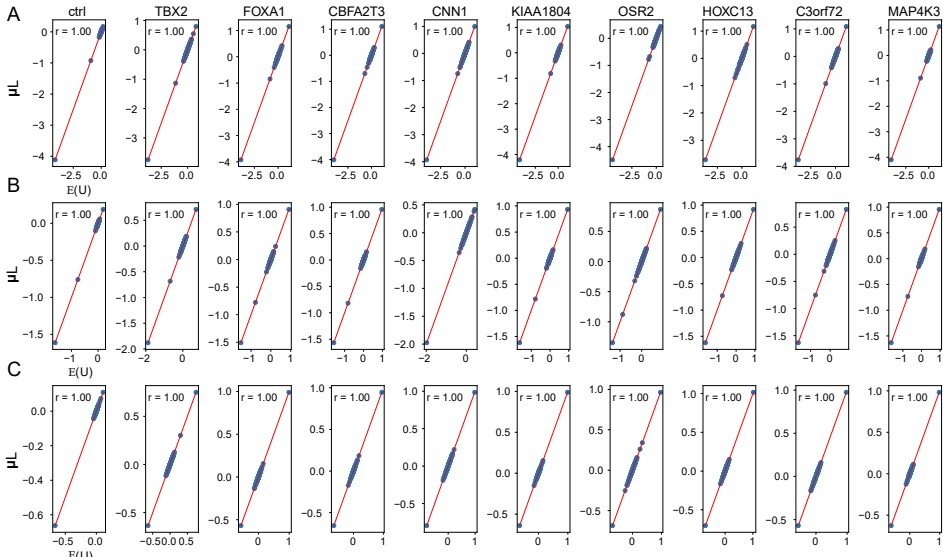

Figure 5: **Experimental validation of derived Eq. 9**. **A-C**. Experimental analysis of derived Eqs. 9 and 10 for the original discrepancy-VAE (MLP) architecture (**A**), **SENA**-$\delta_{\lambda=0.1}$ (**B**) and **SENA**-$\delta_{\lambda=0}$ (**C**), respectively. Here x-axis represent the expected latent factors U extracted by forwarding cells through the trained model for each selected cell type and y-axis represent the (linear) mapping between BP activation scores and pathway archetypes (i..e., latent causal factors).

## II    STUDY ON INFERRED CAUSAL GRAPH ROBUSTNESS

We evaluated how stable are the (directed) edges from the inferred **SENA**-discrepancy-VAE causal graph across various $\lambda$'s and latent dimensions(Fig. 2). To this end, we inferred the causal graph for $\lambda = \{0, 0.1, 10^{-2}, 10^{-3}\}$ and latent dimensions $\{5, 10, 35, 70, 105\}$ using the Norman2019 dataset. We then computed the edge consistency for each graph as the ratio of the frequencies of the most and least frequent sign across the evaluated $\lambda$'s. For instance, if an edge has been consistently positive across $\lambda$'s, it would present an edge consistency of 1 (e.g., Fig. 6-B, $U_{12}$).

When analyzing the edge consistency of the inferred causal graphs (e.g., Fig. 6 A-B, depicted for latent dimension of 5), most edges had a consistency above or equal to 75% while edge weights close to 0 presented low consistency across $\lambda$'s. Interestingly, across the different tested hyperparameters, most edges had a coefficient of variation (CV) $\geq 2$ (Fig. 6 C-F), indicating perfect edge consistency. Importantly, the large majority of edges exhibited a high confidence (CV $\geq 1$)(Fig. 6-H).

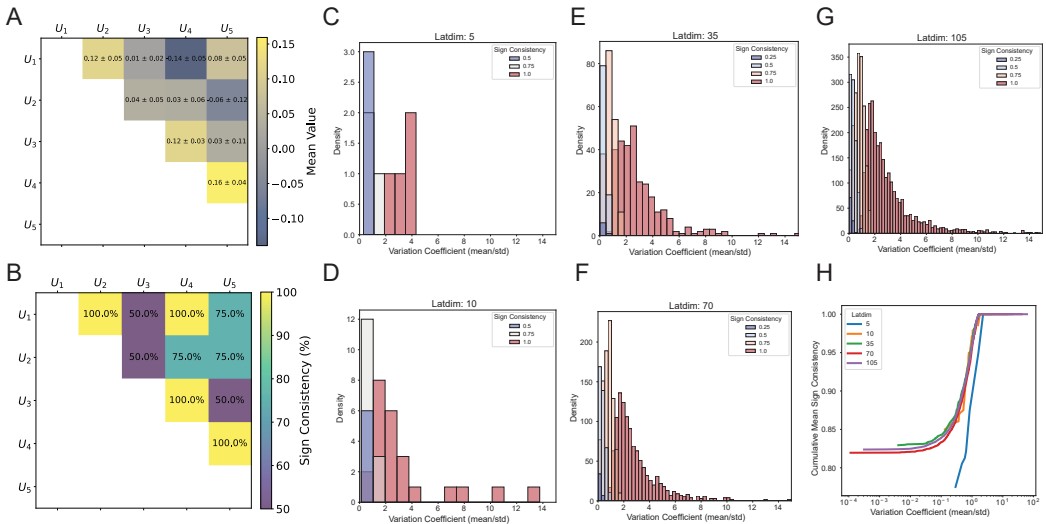

Figure 6: **Edge robustness analysis for the inferred causal graph**. **A**. Edge values (mean of the inferred graph upper triangular. Results are averaged across $\lambda = \{0, 0.1, 10^{-2}, 10^{-3}\}$. **B**. Edge consistency across the aforementioned $\lambda$'s. **C-G**. Coefficient of Variation (mean over standard deviation) histogram for every edge of the upper triangular matrix of the inferred causal graph across several runs using different latent dimensions ($\{5, 10, 35, 70, 105\}$). **H**. Cumulative distribution of the edge consistency.

## III   STUDY ON HIGH-LEVEL-ORDER AGGREGATION OF BIOLOGICAL PATHWAYS

This section analyzes how the connections between the BPs activity after the **SENA** layer and the exogenous variables (meta-pathways) that then define the causal latent factors (pathway archetypes) are potentially biologically-driven, despite the weights being learned in a data-driven way. To this end, we relied on the level 2 Gene Ontology pathways, that encompass most of the used lower-level BPs in our model to evaluate whether the meta-pathway activity encodes high-level biological processes. For the sake of clarity, for this section we would refer to the aforementioned level 2 pathways as L2BPs and keep using BPs to refer to the ones we model at the output of the first layer in **SENA**-discrepancy-VAE.

We selected all L2BPs containing at least 10 BPs (from our set of filtered BPs, i.e. 454 if not stated otherwise), yielding a total of 9 L2BPs with different sizes (Table 2). In order to measure the association between the inferred meta-pathways and the L2BPs, we computed the contribution that each L2BP has on each meta-pathway score and perform a permutation test to evaluate whether the association is significant. We now describe the process for computing these contributions. Be $z_j$ the activation score of the meta-pathway $j$, this activation score is given by the linear combination of previous layer's activation (BPs' activation scores) and the learned weight matrix $\mathbf{\Delta}$ of that layer.

Lets define $\text{L2BP}_k$ as the set of BPs within the $k$-th L2BP. Since the activation score for every meta-pathway $j$ can be expressed as

$$z_j = \sum_{i \in \text{L2BP}_k} \text{BP}_i * \delta_{ij} + \sum_{i' \notin \text{L2BP}_k} \text{BP}_{i'} * \delta_{i'j} \quad \forall k \in \{1, \dots, |\text{L2BP}|\},$$

we can compute the contribution of the $k$-th HLP to the $j$-th latent factor as:

$$c_{kj} = \frac{\sum_{i \in \text{L2BP}_k} \text{BP}_i}{z_j}$$

To measure if this contribution is significant, for a given L2BP and a given meta-pathway, we permuted 1000 times the BPs (maintaining the size) and computed the Mann-Whitney one-sided test to obtain the p-value between permuted and true BPs associated to the L2BP, correcting by the number of tests performed (Bonferroni correction). These allowed us to measure if there is a statistically significant contribution of the specific L2BP (and associated BPs) to the activation score of the given meta-pathway. We performed this analysis on **SENA**-$\delta$ for $\lambda = \{0, 0.1\}$ and, for the sake of simplicity, we set the number of latent dimensions to 35.

Fig. 7 A-B shows the histogram of permuted vs true contributions for every L2BP on the first meta-pathway factor and Fig. 7-C depict the distribution of corrected p-values for every meta-pathway and L2BP, where blanks represent non-significant contributions (corrected p-value $\leq 0.05$). Interestingly, there is a significant contribution for every L2BP across several meta-pathway factors, which may indicate that there are potential clusters of them encoding true high-level biological processes. Also, these results are highly similar across the evaluated $\lambda$ values. Note that this analysis did not discriminate across perturbation types, hence all cells were forwarded (and averaged) through **SENA**'s model to compute the activation scores.

Table 2: Aggregation according to the Level-2 Biological processes of the Gene Ontology structure.

| Level 2 GO Term | #Genesets within | Description |
|---|---|---|
| GO:0022414 | 13 | Reproductive process |
| GO:0002376 | 21 | Immune system process |
| GO:0051179 | 27 | Localization |
| GO:0032501 | 39 | Multicellular organismal process |
| GO:0050896 | 41 | Response to stimulus |
| GO:0008152 | 63 | Metabolic process |
| GO:0032502 | 64 | Developmental process |
| GO:0009987 | 141 | Cellular process |
| GO:0065007 | 193 | Biological regulation |

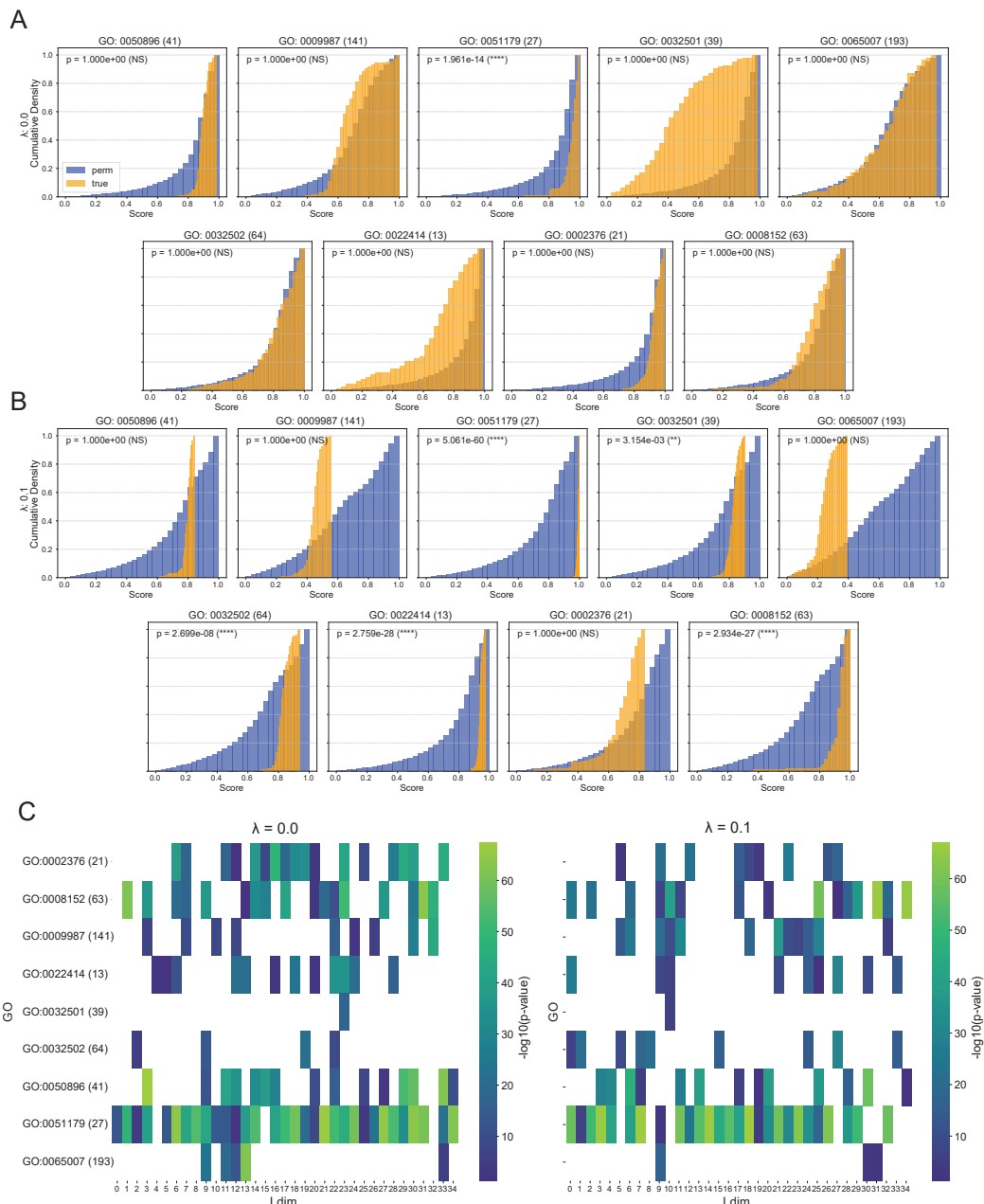

Figure 7: **Study on genesets aggregation at the latent factor level**. **A-B**. Permutation test on the latent factor contribution for level 2 genesets versus random aggregations of genesets, for $\lambda = 0$ (**A**) and $\lambda = 0.1$ (**B**). Results are shown for the first latent factor **C**. Heatmap depicting the corrected p-value for every level 2 geneset and latent factor **SENA**-$\delta$ when $\lambda = 0$ (left) and $\lambda = 0.1$ (right). Number of BPs inside every level 2 GO term are shown in brackets next to the terms' name.

# IV    ABLATION STUDIES ON NORMAN2019'S DATASET

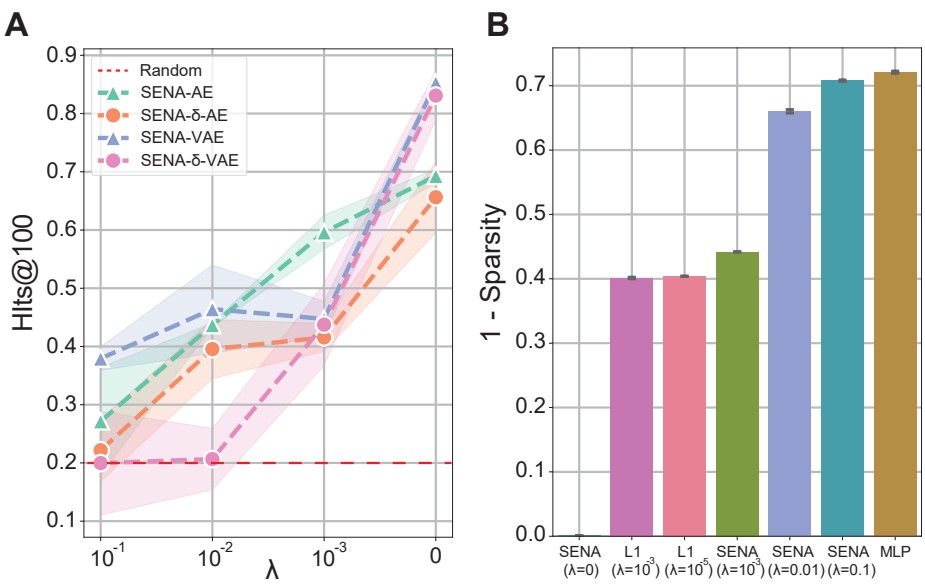

Figure 8: **Ablation studies on interpretability and sparsity**. **A**. Percentage of affected gene sets in the top 100 DA BPs for several **SENA**-based architectures and $\lambda$ values. **B**. Sparsity evaluation according to Eq.**??**.

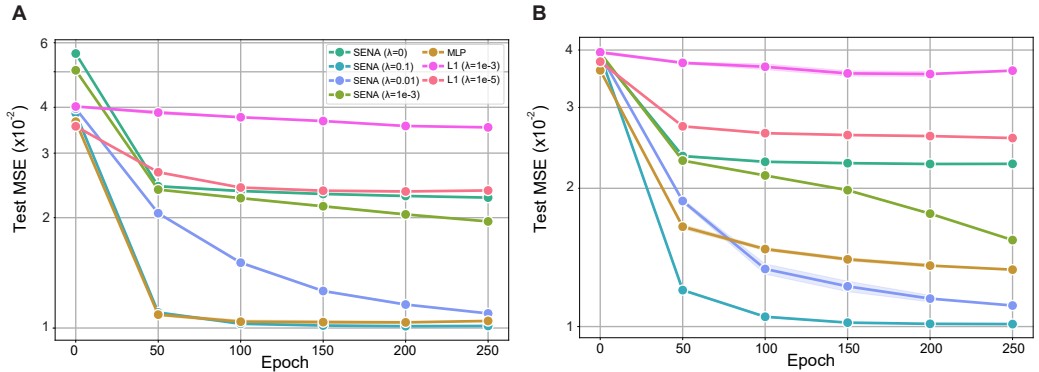

Figure 9: **Ablation studies for AE-type architecture**. Test MSE evaluation for AE-based architectures for **SENA** (**A**) and **SENA**-$\delta$ (**B**) encoders.

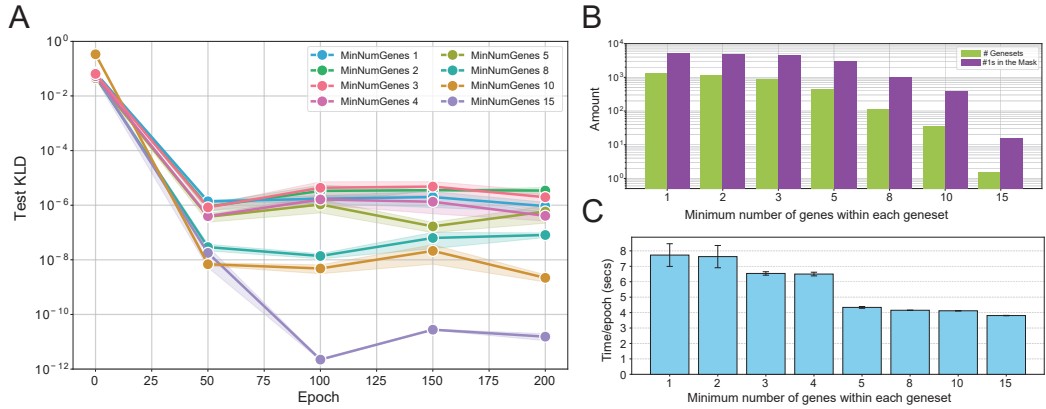

Figure 10: **Ablation studies of the number of BPs used in SENA-**$\delta_{\lambda=0}$. On each experiment, a minimum number of genes per BP is imposed, which reduces the total number of BPs used after the **SENA** layer. **A**. Test KLD as a function of the minimum number of genes within eachBP. **B**. Summary of the number of BPs and the gene-to-BP connections. **C**. Training time-per-epoch results. Results are averaged across 3 different seeds.

Table 3: AE and VAE-based evaluation for **SENA** across 5 seeds. Methods are sorted by sparsity.

| Method | AE-based | VAE-based | |
|---|---|---|---|
| | **Test MSE** ($\times 10^{-2}$) | **Test MSE** ($\times 10^{-2}$) | **Test** $D_{KL}$ ($\times 10^{-4}$) |
| SENA$_{\lambda=0}$ | $2.279 \pm 0.015$ | $3.962 \pm 0.005$ | $13.869 \pm 0.450$ |
| $\ell_{1,\lambda=10^{-3}}$ | $3.526 \pm 0.000$ | $3.954 \pm 0.021$ | $5.016 \pm 4.120$ |
| $\ell_{1,\lambda=10^{-5}}$ | $2.352 \pm 0.028$ | $3.951 \pm 0.005$ | $\mathbf{1.915 \pm 0.127}$ |
| SENA$_{\lambda=10^{-3}}$ | $1.936 \pm 0.023$ | $3.967 \pm 0.023$ | $15.211 \pm 7.351$ |
| SENA$_{\lambda=0.01}$ | $1.103 \pm 0.010$ | $\mathbf{3.938 \pm 0.022}$ | $15.978 \pm 9.905$ |
| SENA$_{\lambda=0.1}$ | $\mathbf{1.009 \pm 0.004}$ | $3.951 \pm 0.009$ | $12.492 \pm 3.054$ |
| MLP | $1.036 \pm 0.012$ | $3.962 \pm 0.019$ | $3.872 \pm 0.225$ |

Table 4: AE and VAE-based evaluation for **SENA**-$\delta$ across 5 seeds. Models are sorted by sparsity.

| Method | AE-based | VAE-based | |
|---|---|---|---|
| | **Test MSE** ($\times 10^{-2}$) | **Test MSE** ($\times 10^{-2}$) | **Test** $D_{KL}$ ($\times 10^{-4}$) |
| **SENA**-$\delta_{\lambda=0}$ | $2.252 \pm 0.011$ | $3.951 \pm 0.012$ | $0.007 \pm 0.009$ |
| $\ell_{1,\lambda=10^{-3}}$ | $3.537 \pm 0.098$ | $\mathbf{3.944 \pm 0.007}$ | $5.742 \pm 3.772$ |
| $\ell_{1,\lambda=10^{-5}}$ | $2.581 \pm 0.011$ | $3.970 \pm 0.006$ | $5.351 \pm 6.089$ |
| **SENA**-$\delta_{\lambda=10^{-3}}$ | $1.552 \pm 0.012$ | $3.973 \pm 0.004$ | $\mathbf{0.001 \pm 0.001}$ |
| **SENA**-$\delta_{\lambda=0.01}$ | $1.096 \pm 0.020$ | $3.969 \pm 0.027$ | $0.024 \pm 0.005$ |
| **SENA**-$\delta_{\lambda=0.1}$ | $\mathbf{1.012 \pm 0.000}$ | $3.966 \pm 0.010$ | $0.326 \pm 0.114$ |
| MLP | $1.350 \pm 0.029$ | $3.954 \pm 0.029$ | $3.816 \pm 1.331$ |

## V    BENCHMARKING ON THE WESSELS DATASET

We included a second large-scale Perturb-seq dataset based on CRISPR-cas13 which aims at efficiently targeting multiple genes for combinatorial perturbations Wessels et al. (2022). This technique, termed CaRPool-seq, encodes multiple perturbations on a cleavable CRISPR array that is associated with a detectable barcode sequence. CaRPool-seq was applied to THP1 cells, an acute myeloid leukemia (AML) model system, to perform combinatorial perturbations of myeloid differentiation regulators and identify their impact on AML differentiation phenotypes.

The perturbations include 28 single perturbations, 26 regulator genes and two negative control genes, as well as 158 double-gene perturbations. We performed standard preprocessing for single cell data (filtering, normalization, and log-transformation), yielding to a total of 424 unperturbed cells (controls), 3036 cells under the 28 single-gene perturbations, and 17592 cells with double-gene perturbations. Pseudo-bulk expression profiles are then obtained by adding gene expression of cells sharing the same perturbation. The resulting profiles are then visualized as UMAP landscape. The same procedure was performed for the Norman2019 dataset (Fig. 11 A and B). Marker genes are then obtained using the Wilcoxon test to contrast the cells from each perturbation with the remaining cells as implemented in Seurat's `FindAllMarkers`, requiring an adjusted p-value < 0.001 and log-fold change > 2.

The Wessels2023 study focused on perturbing myeloid differentiation regulators Wessels et al. (2022). This resulted in all perturbations having similar effects at the transcriptomics levels, as shown in Fig. 11. While in the Norman2019 datasets cells affected by different perturbations tend to cluster separately (panel A), most of the interventions in Wessels2023 are grouped together (panel B), indicating similar profiles. Moreover, the number of genes that are differentially expressed following a perturbation is generally lower in the Wessels2023 study than in Norman2019 (Fig. 11-C), indicating that overall the perturbations in the Wessels2023 data had a more limited effect.

The peculiarities of the Wessels2023 data lead to a notable results: both the **SENA**-discrepancy-VAE and the discrepancy-VAE consistently assign all single-gene perturbations to a single latent factor (figure not shown). This can be interpreted as the models recognizing that all single-gene perturbations have similar effects.

In terms of predictive capabilities (Table 5), we observe that MMD performances on the double perturbations are comparable only when considering higher values of the $\lambda$ parameter. This indicates that for this dataset, introducing interpretability in addition to representational capabilities is more difficult, possibly due to the peculiarities of the performed experiments.

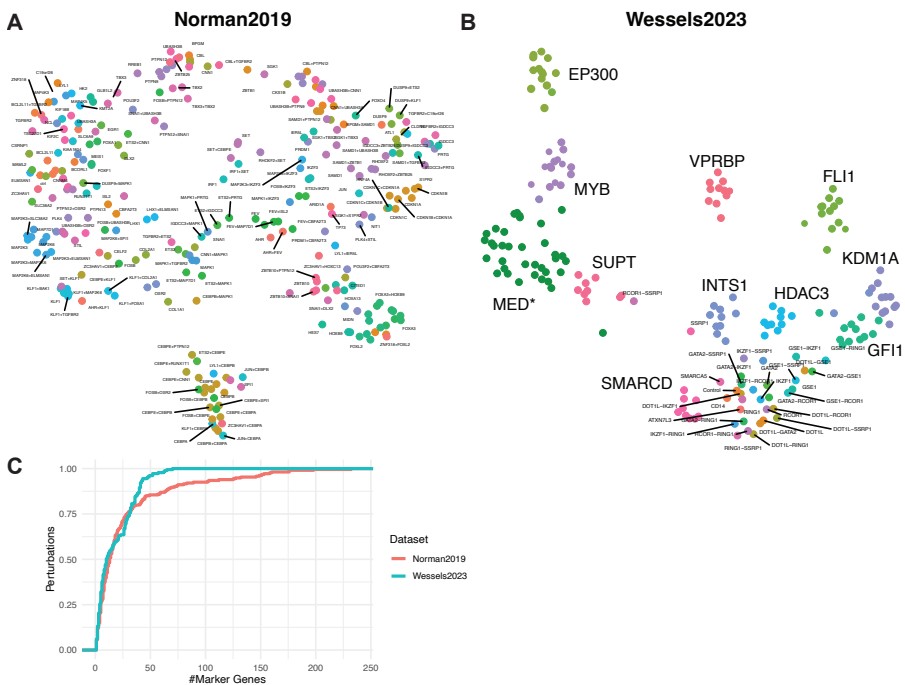

Figure 11: **Comparison of Norman and Wessels datasets**. **A**. UMAP representation of single-gene and combinatorial perturbations captured in the Norman2019 dataset. Each point represents the pseudo-bulk expression profile of a genetic perturbation. **B**. UMAP representation of the Wessels2023 dataset similar to A. Perturbations which share one gene and form clear clusters are shown in the same color to enhance clarity, while the remaining perturbations are shown individually.**C**. Cumulative density distribution of perturbations by their respective number of marker genes (differentially expressed genes with $p_{adj}$<0.001, average LFC > 2) per perturbation in the Norman2019 and Wessels2023 dataset.

Table 5: Performance comparison between SENA-discrepancy-VAE and discrepancy-VAE on the Wessel2023 dataset across different lambda values and latent factors for double perturbation samples. Note that KLD and L1 losses are not dependent on the samples, but computed after the training process is finished.

| Encoder | Metric | Latent Dimension | | |
|---|---|---|---|---|
| | | **50** | **28** | **14** |
| Original MLP | MMD↓ | **0.1810** $\pm$ 0.0001898 | **0.1768** $\pm$ 0.0001898 | 0.1886 $\pm$ 0.0001225 |
| | MSE↓ | **0.0767** $\pm$ 0.0000031 | **0.0840** $\pm$ 0.0000031 | 0.0934 $\pm$ 0.0000005 |
| | KLD↓ | 0.0137 $\pm$ 0.0000011 | 0.0142 $\pm$ 0.0000011 | 0.0150 $\pm$ 0.0000002 |
| | L1↓ | 0.0054 $\pm$ 0.0000001 | 0.0028 $\pm$ 0.0000001 | 0.0016 $\pm$ 0.00000004 |
| SENA-$\delta_{\lambda=0.5}$ | MMD↓ | 0.2285 $\pm$ 0.0011976 | 0.1904 $\pm$ 0.0001348 | **0.1753** $\pm$ 0.0000720 |
| | MSE↓ | 0.0934 $\pm$ 0.0000011 | 0.0849 $\pm$ 0.0000108 | **0.0782** $\pm$ 0.0000003 |
| | KLD↓ | 0.0130 $\pm$ 0.0000002 | 0.0135 $\pm$ 0.0000025 | 0.0107 $\pm$ 0.0000003 |
| | L1↓ | **0.0013** $\pm$ 0.0000001 | 0.0021 $\pm$ 0.0000018 | 0.0052 $\pm$ 0.0000002 |
| SENA-$\delta_{\lambda=0.1}$ | MMD↓ | 0.3637 $\pm$ 0.0024426 | 0.4267 $\pm$ 0.0027825 | 0.4049 $\pm$ 0.0038001 |
| | MSE↓ | 0.1002 $\pm$ 0.0000042 | 0.1081 $\pm$ 0.0000056 | 0.1086 $\pm$ 0.0000008 |
| | KLD↓ | 0.0040 $\pm$ 0.0000001 | 0.0049 $\pm$ 0.0000001 | 0.0066 $\pm$ 0.0000015 |
| | L1↓ | 0.0090 $\pm$ 0.0000021 | **0.0026** $\pm$ 0.0000001 | **0.0013** $\pm$ 0.00000001 |
| SENA-$\delta_{\lambda=0}$ | MMD↓ | 0.5675 $\pm$ 0.0031239 | 0.6156 $\pm$ 0.0041942 | 0.5725 $\pm$ 0.0096761 |
| | MSE↓ | 0.1108 $\pm$ 0.0000037 | 0.1068 $\pm$ 0.0000076 | 0.1131 $\pm$ 0.0000011 |
| | KLD↓ | **0.0019** $\pm$ 0.00000004 | **0.0025** $\pm$ 0.0000001 | **0.0027** $\pm$ 0.0000001 |
| | L1↓ | 0.0120 $\pm$ 0.0000014 | 0.0040 $\pm$ 0.0000010 | 0.0014 $\pm$ 0.0000001 |

## VI   EVALUATING GEARS ON PREDICTING UNSEEN DOUBLE PERTURBATIONS

We trained GEARS, following the authors' recommendations, for 20 epochs on all single-gene perturbations from the Norman2019 dataset, and predicted the same double-perturbations we used to evaluate **SENA**-discrepancy-VAE and the original discrepancy-VAE. We repeated this experiment with 5 different initializations.

We then computed the MMD and MSE between the predicted and the true double-perturbations gene expression profiles by forwarding randomly-selected unperturbed cells (control) and averaged the metrics across perturbations. We show the MMD in the main benchmarking (Table 1) and in this section's Table 6 to compare against MSE. When analyzing these results, we found that the MSE on double perturbations exhibited scores 10 times larger than the one reported during training for validation folds ($0.00368 \pm 0.000363$ across different latent dimension sizes for one seed), which suggest a potential lack of generalization. Moreover, MMD also showed significantly larger scores compared to **SENA** or the standard discrepancy-VAE. This may be justified by the fact that the MMD is not used as a loss function in GEARS, hence true and predicted distributions may differ significantly when MSE is large enough. Finally, in order to have a baseline within GEARS, we computed the MSE and MMD between the true expression of double perturbations and true expression of unperturbed cells, yielding an average value of MMD = 14.39 and MSE = 0.085.

Overall, these results highlight that **i)** the hidden dimension size of GEARS is largely dependent of the prediction performance. **ii)** GEARS outperforms the defined baseline in terms of MSE for some range of latent dimensions. **iii)** GEARS fails to capture the underlying distribution of double perturbations, possibly due to the lack of a distribution-distance loss function during training.

Table 6: MMD and MSE of GEARS on Norman2019's dataset for double perturbation prediction across several latent dimensions

| Metric | Latent Dimension | | | | |
|---|---|---|---|---|---|
| | **105** | **70** | **35** | **10** | **5** |
| MMD↓ | $14.9420 \pm 0.233957$ | $12.6036 \pm 0.745302$ | $13.2590 \pm 0.559124$ | $12.9774 \pm 0.751373$ | $15.3099 \pm 0.191833$ |
| MSE↓ | $0.0986 \pm 0.009157$ | $0.0446 \pm 0.018703$ | $0.0562 \pm 0.015065$ | $0.0533 \pm 0.019782$ | $0.1131 \pm 0.008568$ |

## VII  ADDITIONAL MATERIALS ON NORMAN2019 DATASET

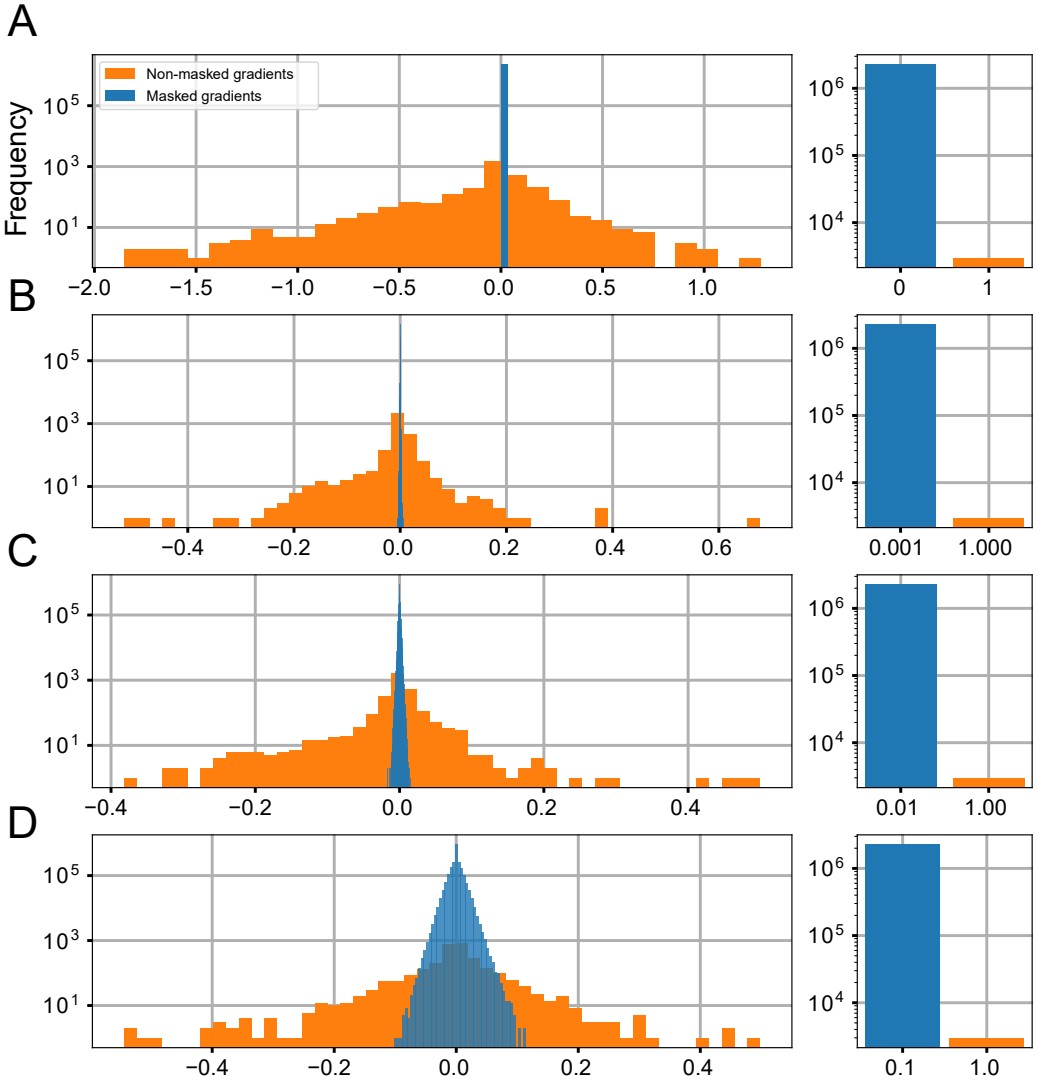

Figure 12: **SENA layer analysis**. Masked and non-masked gradients for **SENA**-discrepancy-VAE at the output of the **SENA** layer, for $\lambda$ values of 0 (**A**), $10^{-3}$ (**B**), 0.01 (**C**) and 0.1 (**D**). Barplot showing matrix $M$ values is depicted next to each histogram.

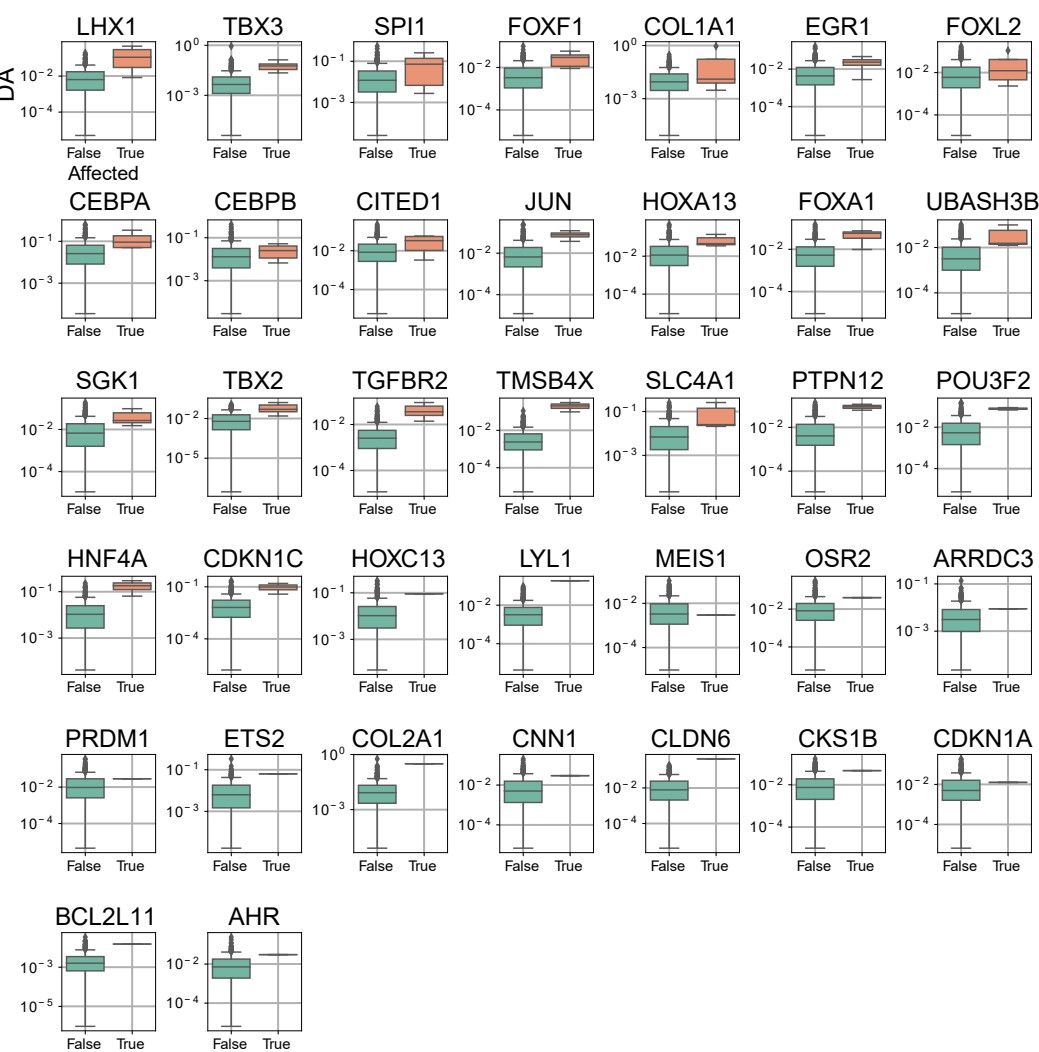

Figure 13: **Differential Activation Scores analysis**. DA score analysis for targeted and non-affected BPs along the 37 single-gene perturbations present in the input gene expression matrix.

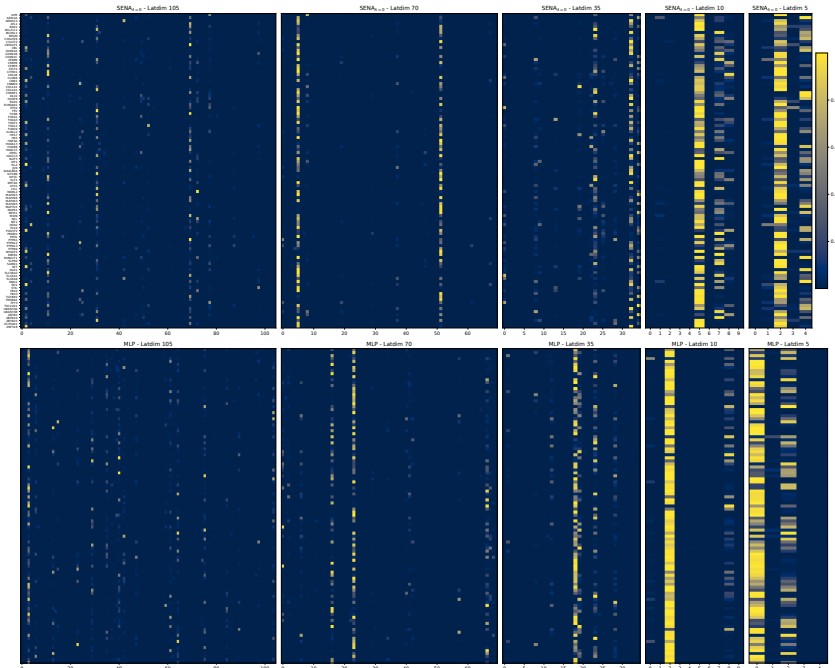

Figure 14: **Mapping between latent factors and perturbations genes**. Mapping distribution of knocked out genes and latent factors, from **SENA**-discrepancy-VAE ($\lambda = 0$) and discrepancy-VAE (MLP encoder) for several values of latent dimensions {105, 70, 35, 10, 5}. We generated this mapping from the interventional encoder according to $h = \mathrm{Softmax}\left(\mathrm{Linear}\left(\mathrm{LeakyReLU}\left(\mathrm{Linear}(c)\right)\right) \times temp\right)$, where c is the one-hot encoding vector for each perturbation and $temp$ was set to 100 as recommended for inference in the original manuscript.

Table 7: **Causal graph (Fig. 2) details**. Each row lists a latent factor, the number of targeted perturbations, and associated biological processes within it.

| Latent Factor | Targeting Perturbations | Biological Processes |
|---|---|---|
| 41 | 41 | 57 |
| 65 | 6 | 10 |
| 2 | 18 | 14 |
| 53 | 25 | 10 |
| 69 | 1 | 1 |
| 15 | 1 | 1 |
| 12 | 9 | 10 |

Table 8: **Top 6 (knockout, gene set) pairs according to DA score. Fig. 4 details)**. DA scores for selected genes and associated GO terms.

| Gene | GO Term | DA Score |
|---|---|---|
| COL1A1 | GO:0038065 | 0.92 |
| TBX3 | GO:0006833 | 0.88 |
| GLB1L2 | GO:0070141 | 0.71 |
| TP73 | GO:0006833 | 0.69 |
| CEBPA | GO:0010714 | 0.65 |
| CEBPA | GO:0006595 | 0.60 |

Table 9: Mapping between BP and latent factors.

| Latent Factor | GO ID | GO Term |
|---|---|---|
| 2 | GO:2000352 | negative regulation of endothelial cell apoptotic process |
| 2 | GO:0010944 | negative regulation of transcription by competitive promoter binding |
| 2 | GO:1904292 | regulation of ERAD pathway |
| 2 | GO:0060216 | definitive hemopoiesis |
| 2 | GO:0048557 | embryonic digestive tract morphogenesis |
| 2 | GO:0071549 | cellular response to dexamethasone stimulus |
| 2 | GO:0055023 | positive regulation of cardiac muscle tissue growth |
| 2 | GO:0071243 | cellular response to arsenic-containing substance |
| 2 | GO:0006067 | ethanol metabolic process |
| 2 | GO:0099188 | postsynaptic cytoskeleton organization |
| 2 | GO:0006833 | water transport |
| 2 | GO:0045723 | positive regulation of fatty acid biosynthetic process |
| 2 | GO:2001279 | regulation of unsaturated fatty acid biosynthetic process |
| 2 | GO:0018904 | ether metabolic process |
| 12 | GO:0070141 | response to UV-A |
| 12 | GO:0060512 | prostate gland morphogenesis |
| 12 | GO:1903902 | positive regulation of viral life cycle |
| 12 | GO:0014821 | phasic smooth muscle contraction |
| 12 | GO:0006359 | regulation of transcription by RNA polymerase III |
| 12 | GO:0030852 | regulation of granulocyte differentiation |
| 12 | GO:0044849 | estrous cycle |
| 12 | GO:0045737 | positive regulation of cyclin-dependent protein serine/threonine kinase activity |
| 12 | GO:1903319 | positive regulation of protein maturation |
| 12 | GO:2000010 | positive regulation of protein localization to cell surface |
| 15 | GO:0050665 | hydrogen peroxide biosynthetic process |
| 41 | GO:0010226 | response to lithium ion |
| 41 | GO:0002693 | positive regulation of cellular extravasation |
| 41 | GO:0045654 | positive regulation of megakaryocyte differentiation |
| 41 | GO:0060065 | uterus development |
| 41 | GO:1902042 | negative regulation of extrinsic apoptotic signaling pathway via death domain receptors |
| 41 | GO:0000305 | response to oxygen radical |
| 41 | GO:0001915 | negative regulation of T cell mediated cytotoxicity |
| 41 | GO:0002281 | macrophage activation involved in immune response |
| 41 | GO:0002357 | defense response to tumor cell |
| 41 | GO:0006595 | polyamine metabolic process |
| 41 | GO:0006921 | cellular component disassembly involved in execution phase of apoptosis |
| 41 | GO:0006957 | complement activation, alternative pathway |
| 41 | GO:0010714 | positive regulation of collagen metabolic process |
| 41 | GO:0010829 | negative regulation of glucose transmembrane transport |
| 41 | GO:0014912 | negative regulation of smooth muscle cell migration |
| 41 | GO:0019835 | cytolysis |
| 41 | GO:0030449 | regulation of complement activation |
| 41 | GO:0032703 | negative regulation of interleukin-2 production |
| 41 | GO:0032905 | transforming growth factor beta1 production |
| 41 | GO:0033005 | positive regulation of mast cell activation |
| 41 | GO:0039532 | negative regulation of cytoplasmic pattern recognition receptor signaling pathway |
| 41 | GO:0044342 | type B pancreatic cell proliferation |
| 41 | GO:0044406 | adhesion of symbiont to host |
| 41 | GO:0045056 | transcytosis |
| 41 | GO:0045663 | positive regulation of myoblast differentiation |
| 41 | GO:0050849 | negative regulation of calcium-mediated signaling |
| 41 | GO:0051764 | actin crosslink formation |
| 41 | GO:0061450 | trophoblast cell migration |
| 41 | GO:0070102 | interleukin-6-mediated signaling pathway |
| 41 | GO:0070486 | leukocyte aggregation |
| 41 | GO:0071391 | cellular response to estrogen stimulus |

Table 10: Mapping between BP and latent factors (Cont.).

| Latent Factor | GO ID | GO Term |
|---|---|---|
| 41 | GO:0097202 | activation of cysteine-type endopeptidase activity |
| 41 | GO:0099515 | actin filament-based transport |
| 41 | GO:0140374 | antiviral innate immune response |
| 41 | GO:1900120 | regulation of receptor binding |
| 41 | GO:1900745 | positive regulation of p38MAPK cascade |
| 41 | GO:1902307 | positive regulation of sodium ion transmembrane transport |
| 41 | GO:2000508 | regulation of dendritic cell chemotaxis |
| 41 | GO:0020027 | hemoglobin metabolic process |
| 41 | GO:0002227 | innate immune response in mucosa |
| 41 | GO:0034505 | tooth mineralization |
| 41 | GO:0038065 | collagen-activated signaling pathway |
| 41 | GO:0001958 | endochondral ossification |
| 41 | GO:0034389 | lipid droplet organization |
| 41 | GO:0010715 | regulation of extracellular matrix disassembly |
| 41 | GO:0051131 | chaperone-mediated protein complex assembly |
| 41 | GO:0060384 | innervation |
| 41 | GO:0061684 | chaperone-mediated autophagy |
| 41 | GO:0055091 | phospholipid homeostasis |
| 41 | GO:0060742 | epithelial cell differentiation involved in prostate gland development |
| 41 | GO:0097066 | response to thyroid hormone |
| 41 | GO:0042976 | activation of Janus kinase activity |
| 41 | GO:0061323 | cell proliferation involved in heart morphogenesis |
| 41 | GO:0002713 | negative regulation of B cell mediated immunity |
| 41 | GO:0051238 | sequestering of metal ion |
| 41 | GO:0062098 | regulation of programmed necrotic cell death |
| 41 | GO:2000479 | regulation of cAMP-dependent protein kinase activity |
| 53 | GO:0016338 | calcium-independent cell-cell adhesion via plasma membrane cell-adhesion molecules |
| 53 | GO:0009713 | catechol-containing compound biosynthetic process |
| 53 | GO:0032060 | bleb assembly |
| 53 | GO:0036035 | osteoclast development |
| 53 | GO:0007190 | activation of adenylate cyclase activity |
| 53 | GO:0009065 | glutamine family amino acid catabolic process |
| 53 | GO:0035767 | endothelial cell chemotaxis |
| 53 | GO:2000678 | negative regulation of transcription regulatory region DNA binding |
| 53 | GO:0043032 | positive regulation of macrophage activation |
| 53 | GO:0060749 | mammary gland alveolus development |
| 65 | GO:0021516 | dorsal spinal cord development |
| 65 | GO:0017014 | protein nitrosylation |
| 65 | GO:0010842 | retina layer formation |
| 65 | GO:0030540 | female genitalia development |
| 65 | GO:0042759 | long-chain fatty acid biosynthetic process |
| 65 | GO:2000696 | regulation of epithelial cell differentiation involved in kidney development |
| 65 | GO:0015701 | bicarbonate transport |
| 65 | GO:0034638 | phosphatidylcholine catabolic process |
| 65 | GO:0048535 | lymph node development |
| 65 | GO:0050995 | negative regulation of lipid catabolic process |
| 69 | GO:0001886 | endothelial cell morphogenesis |

