# OpenReview forum: "Interpretable Causal Representation Learning for Biological Data in the Pathway Space"
_ICLR.cc/2025/Conference — ICLR 2025 Poster_

### Official Review · Reviewer_ox26 · 2024-11-02

**Soundness:** 3
**Presentation:** 3
**Contribution:** 3
**Rating:** 6
**Confidence:** 4

**Summary:**

This paper introduces SENA-discrepancy-VAE, a novel model in causal representation learning (CRL) designed to make biological data analysis—especially from Perturb-seq experiments—more interpretable. A key innovation of this work is how it integrates biological processes (BPs) as prior knowledge, directly linking the model’s latent factors to known biological pathways. This approach fills an important gap in existing CRL methods, which often struggle with interpretability since they don't directly associate learned representations with actual biological mechanisms, making them less useful for real research applications.

SENA-discrepancy-VAE builds on the standard discrepancy-VAE by introducing a pathway-based masking strategy within a new encoder, SENA-δ. This encoder uses a two-layer masked MLP where the first layer maps gene expression values to BP activity levels, with a tunable parameter that adjusts the influence of genes outside predefined pathways, giving the model flexibility in gene-pathway associations. The second layer models latent factors as combinations of these BP activities, which is a more realistic approach since biological interventions often impact multiple pathways. This setup stays true to the CRL assumption that each intervention targets a single latent factor but does so in a way that aligns with biological realities.

The authors evaluate SENA-discrepancy-VAE on a Perturb-seq dataset of leukemia lymphoblast cells. They show that the model performs as well as the original discrepancy-VAE on unseen perturbation combinations while providing greater interpretability by identifying specific BPs associated with each latent factor. This interpretability is validated through pathway-specific analysis, demonstrating the model’s ability to reveal biologically meaningful patterns in response to genetic interventions.

**Strengths:**

1) The proposed model’s integration of biological pathway data as a prior in the causal representation learning (CRL) framework is interesting and practical, offering a biologically grounded solution to gene expression analysis. Through the direct alignment of the latent factors with biological pathways, SENA-discrepancy-VAE addresses the common limitation in CRL models of producing uninterpretable latent factors.

2) The paper presents thorough experiments across multiple perturbation types, including single and double-gene knockouts, demonstrating the model’s robustness. Its generalization to unseen perturbations is compelling, and the ablation studies, which explore interpretability-reconstruction trade-offs, further validate the model’s design.

3) The authors have done a good job communicating the importance of embedding biological processes into latent spaces, with visuals that illustrate BP-specific activity levels influencing latent factors. The use of causal graphs and the differential activation (DA) metric enhances transparency, allowing readers to trace latent factors back to biological functions.

4) The model’s ability to provide interpretable predictions on cellular responses can aid in experimental design and offer insights into the potential effects of genetic or drug interventions. This approach addresses a pressing need in biomedicine for interpretable causal representation learning (CRL) models that can shed light on the intricate causal relationships underlying gene function and cellular processes.

**Weaknesses:**

1) While the model effectively identifies single-point perturbations, it does not accommodate multi-step perturbations or capture the progression of cellular responses over time. In biological experiments, the cellular response often evolves in phases, with gene activity showing distinct transitions that are essential for understanding the effect of interventions.

2) The model’s validation on a single dataset (K562 cell line data) restricts insights into its generalizability across different cell types or conditions. Testing on additional datasets, such as those from other cell lines or cellular environments, would offer a stronger assessment of robustness and applicability across a wider range of biological data.

3) The model assumes static pathway relevance across all tasks, which may limit its adaptability in varied biological contexts where pathway importance changes with cell type or condition.

4) The paper assumes that each intervention corresponds to a single latent factor, limiting the model's ability to capture complex interactions where multiple latent factors might be influenced by a single intervention. This simplification restricts the model’s interpretability in representing overlapping or interacting biological processes, which are common in gene expression dynamics.

5) The model lacks a detailed exploration of how varying the number of latent dimensions impacts the interpretability and causal mapping of latent factors. Larger or smaller dimensions can influence the granularity of the factors and thus affect the biological insights the model can provide.

**Questions:**

1) Could the authors clarify whether multi-label or probabilistic pathway activity labels were considered as an alternative to binary labels? Binary labels may oversimplify gene activity levels, especially when certain pathways exhibit gradations rather than discrete on/off states.

2) How would the model perform if pathway relevance were dynamically adjusted based on specific task contexts or experimental conditions? Pathway importance often varies, and adapting pathway relevance could improve model flexibility.  A task-specific analysis to explore whether dynamically adjusting pathway selection enhances generalizability across datasets and biological contexts could provide insights into the model’s robustness.

3) In the causal graph (Figure 3), how sensitive are the edge weights to the choice of λ and latent dimension? Please provide a sensitivity analysis.

4) The mask matrix M (equation 2) assumes binary gene-pathway relationships. Have you considered using weighted relationships based on pathway databases' confidence scores?

5) How does D_KL evolve during training for different λ values? Training curves would help understand if this is an optimization or regularization effect. Does this phenomenon persist if you randomize the pathway annotations while maintaining the same sparsity structure? This would determine if the benefit comes from biological knowledge or just sparsity.

6) Have the authors empirically validated that the expectation in Equation (10) aligns with the observed experimental outcomes? Demonstrating this match would reinforce the theoretical assumptions and provide additional confidence in the model's causal interpretability.

---

> ### Author Response · Authors · 2024-11-22
>
> **Weaknesses:**
> 1. We thank the reviewer for this interesting proposal. Indeed, we have not tested the proposed model for multiple timeframes perturbations, although we believe it could benefit from imposing temporal constraints (e.g. Peng et al. Communications Biology (2023)) over the data, which is not natively supported in our method nor the standard discrepancy-VAE. However, we plan to address this issue in future work.
> 2. We thank the reviewer for highlighting this limitation. The updated manuscript has incorporated a second dataset, Wessel2023, which is based on a different knockout technology, CRISPR-cas13, and can further validate the interpretability and performance of the proposed model. Also, we are planning to include a third dataset to strengthen its generalizability.
> 3. We thank the reviewer for this logical concern. Although it is true that we are proposing a mask for the SENA layer which imposes fixed pathway relevance, the weights assigned to the relationship between input genes and gene sets within the SENA layer are not fixed, which allow learning dynamic pathway relevance along the training process. Hence, this can adapt to different cell types and conditions. We acknowledge that this may be not clearly explained in the text, and we have further underscore these differences in the process where the binary mask is applied to the weights.
> 4. This is an extremely interesting point of discussion. The assumption that each intervention targets a single latent factor is one of the cornerstones of the causal representation learning framework first introduced by Ahuja et al. [https://proceedings.mlr.press/v202/ahuja23a.html] and upon which both the discrepancy-VAE and our SENA-discrepancy-VAE models are based. Without this assumption, the identifiability of the causal latent factors is not guaranteed.
> A practical consequence of this assumption is that if an intervention targets multiple latent factors, then it is not possible  to disentangle these latent factors, and they will be collapsed into one.
> Furthermore, for the SENA-discrepancy-VAE model, this means that if two interventions target overlapping sets of biological processes, then both interventions will be associated with the same latent factor, with the latter in turn incorporating all gene sets targeted by the two interventions. This extends to the case of n interventions as well. And this explains why we observe only few latent factors being targeted by interventions, both in the discrepancy-VAE and in the SENA-discrepancy-VAE. This issue is indicated in our discussion as well, where we remark that future research in CRL methods should try to overcome the assumption of one single latent factor targeted by each intervention.
> 5. We thank the reviewer for expressing this concern. Along the paper, we have delved into this kind of analysis on two occasions. 1) We have analyzed the DAR metric, which evaluates the interpretability of a perturbation across genesets, and showed that it presented stable results across several numbers of latent dimensions, yielding robust biological insights. 2) We apologize for the typo of not referring to Figure 8 on the conclusion of the paper (and instead referring to Table 2), which explains the mappings between perturbations, according to the categorical encoder of the standard discrepancy-VAE, across a number of latent factors. Nevertheless, we plan to further expand interpretability analysis at the output of the SENA layer to include several latent dimensions. We will present the results of this experiment in the final version of the manuscript.

---

> > ### Author Response · Authors · 2024-11-22
> >
> > **Questions:**
> > 1. Although it is true that we are proposing a binary mask for the SENA layer, we believe it is not a limiting factor for two reasons: 1) the genes that we considered targeted gene sets have a Lambda=1, which present the same freedom any weight in the network has, hence they can vary freely along the training process according to the established loss function. 2) the remaining genes (those that are not targeting specific gene sets) will depend on the lambda value, but again are even more heavily dependent on the matrix’s weight they acquired during the training process. Moreover, since we have already empirically validated in Ruiz et al, 2024 (Results, section 3) that an autoencoder-based model trained with the interpretable SENA layer we are proposing is able to learn the specific contributions each gene has over each geneset in a biologically-meaningful manner.
> > 2. We thank the reviewer for this interesting question. We believe that making the λ parameter learnable, for instance in an attention-based manner, could indeed dynamically adjust the pathway relevance to specific tasks contexts. We are incorporating this experiment into the final version of the manuscript.
> > 3. We thank the reviewer for the interesting proposal. We are planning to incorporate a small ablation study on the causal graph generation to evaluate sensitivity of edge weights as a function of λ and latent dimension.
> > 4. We thank the reviewer for the interesting question. Although these confidence scores would be an interesting experiment to perform, we are not aware of any existing confidence scores for gene/gene sets that we can use. Moreover, similar to what we have expressed above, the assumed binary gene-pathway relationship does not limit performance nor interpretability. Also, will include an analysis where lambda is considered a learnable parameter, adding an extra layer of freedom to SENA.
> > 5. We will extend the ablation studies to evaluate D_KL as a function of λ, and incorporate the randomization of pathway annotation experiments into them.
> > 6. We thank the reviewer for his concern on providing robust theoretical results to the proposed approach. We aim at including this experiment on the final version of the manuscript.

---

> > > ### Author Response · Authors · 2024-11-28
> > >
> > > We sincerely thank the reviewer once again for the valuable suggestions and comments. In the updated manuscript, we have incorporated several experiments to address the concerns raised.
> > >
> > > First, we included a second dataset, Wessels2023, compared it with Norman2019 in terms of perturbations and transcriptomic profiles, and evaluated it using both the standard and SENA discrepancy-VAEs. This analysis is detailed in the first comment’s update and further explored in Appendix V.
> > >
> > > Additionally, we conducted a robustness analysis of the inferred causal graph across various latent dimensions and λ values, highlighting the reliability of the uncovered causal mechanisms (see Appendix II). Furthermore, we performed an experiment to investigate the dynamic relevance of BPs across latent factors, revealing patterns resembling higher-order GO pathways (Appendix III).
> > >
> > > We believe these additions effectively address the reviewer’s concerns regarding the limited datasets, the impact of latent dimensions on factor granularity, and the static nature of pathway relevance. These points are described in detail in the first comment’s update.
> > >
> > > As always, we are happy to clarify or address any further questions regarding this revised version of the manuscript.

---

> > > > ### Comment · Reviewer_ox26 · 2024-11-28
> > > >
> > > > Thanks for your comments. I  have reviewed the additions made and these meet the requirements I had stated. I will increase my score to reflect the same.

---

### Official Review · Reviewer_ez4D · 2024-11-03

**Soundness:** 2
**Presentation:** 3
**Contribution:** 3
**Rating:** 6
**Confidence:** 3

**Summary:**

This paper presents SENA-discrepancy-VAE, an extension of the discrepancy-VAE framework that incorporates biological pathway knowledge to produce interpretable causal latent factors. The authors modify the encoder architecture to map gene expression through biological processes (BPs) while maintaining the theoretical guarantees of the original model. The approach achieves comparable predictive performance to the original discrepancy-VAE while providing biologically meaningful latent representations and interpretability.

**Strengths:**

* Clarity: The paper is well written and easy to follow.

* Novel Technical Contribution: The paper successfully extends causal representation learning to incorporate domain knowledge while preserving theoretical guarantees. The SENA-δ encoder architecture is a clever solution to balance interpretability and performance.

* Practical Impact: The work addresses a significant gap in current causal representation learning methods for biological data, where interpretability is crucial for scientific insights. The ability to map latent factors to biological processes makes the model more useful for domain experts.

**Weaknesses:**

* Limited Biological Validation: While the authors show statistical associations between perturbations and biological processes, there could be more validation using external biological knowledge or experimental validation of the discovered causal relationships.

* Hyperparameter Sensitivity: The model introduces an additional hyperparameter λ that significantly impacts performance. While ablation studies are provided, more guidance on selecting this parameter would be valuable (this is important given that there's some large impact on the performance of the method)

* Restricted Evaluation: The empirical evaluation is limited to a single dataset (Norman et al., 2019). Additional validation on different types of biological data would strengthen the claims of generalizability.

**Questions:**

Can you provide examples of BPs in the appendix?

Regarding DAR evaluation: What happens if unnaffected pathways have very low action? Also, more general, how would you deal with impalanced pathways, which might lead in measuring large noise levels?

In table 2, for SENA λ=0.1, latent dim 105, the variance compared to original MLP and λ=0 is significantly lower (0.000081 vs 0.001087). Just double checking if this is corrent.

L237: During filtering you end up with a (biased) set of BPs. How much do you think this can influence interpretability? Is there a risk of removing useful BPs?

Suggestions
L100: the word faithfully here gets confused with causal faithfulness. Please consider an alternative adverb if possible.
L105: target instead of targets (remove final s)

---

> ### Author Response · Authors · 2024-11-22
>
> **Weaknesses:**
> 1. We thank the reviewer for the interesting question, and we acknowledge the limited analysis on this topic. Reviewer 1 also pointed this out. To address this issue, we are working on incorporating experiments treating the biological plausibility as how the latent factors maintain the biological structure of the GO terms in higher aggregation levels. Additionally, we are currently working on delving into the biological plausibility of the inferred causal graph. We hope to provide an in-depth analysis in the next iteration of the manuscript.
> 2. We completely agree with the reviewer in this matter and we consider it important to provide some sweet-spot boundaries of lambda where SENA can get both interpretable and efficient results across datasets. Due to this, we are extending the ablation studies on the lambda evaluation for the incorporated datasets and we will present the results on the final version of the manuscript.
> 3. We thank the reviewer for highlighting this limitation. The updated manuscript has incorporated a second dataset, Wessels2023, which is based on a different knockout technology, CRISPR-cas13, and can further validate the interpretability and performance of the proposed model. Also, we are planning to include a third dataset to strengthen its generalizability.
>
> **Questions:**
> 1. We apologize for not referencing in the text Table 6 and Table 7, which contain the mapping between BPs and latent factors of the causal graph in Figure 2. These provide the GO id and description for each BP. We have now referenced this in the text.
> 2. We thank the reviewer for the interesting question. Since DA is reflecting the ratio over perturbed and control cells expression on a specific geneset (that is, a neuron of the NN), the effect of low action genesets will be smoothed by this ratio, since it will affect both the perturbed and control cells. Importantly, we only include gene sets containing at least 5 or more genes measured in our dataset, so as to remove smaller noisy sets.  Moreover, in a similar context, when averaging DAs to compute DAR, even though it is true that some DAs would be affected by only a few genes (a minimum of 5 is enforced) or a greater number, the average on the DAR will smoothed the effect of noise levels, underscoring the robustness of this metric. We believe this matter was not clearly explained in the manuscript, hence we have clarified it (Section 4.3, final paragraph).
> 3. We thank the reviewer for spotting this concern. Indeed, these values are correct, however we are increasing the number of seeds for this table, from 3 to 5, to smooth out the differences in the presented results.
> 4. There are 3 main filterings that could “bias” the used set of genesets: 1) We filtered those gene sets containing >= 50% in common with other genesets. 2) We removed gene sets that introduced unstabled and unreplicable results, following Ruiz-Arenas et al., Nucleic Acid Research (2024) practices. 3) We filtered those genesets containing < 5 genes. Although it is true that this can indeed remove some interesting genesets, we believe these conditions ensure that results are robust across runs, and present genesets reflect independent activity, due to the low level of intersection with other genesets. Moreover, these conditions are preprocessing parameters that can be modified prior to running SENA. We have clarified this in the text and included these options in the SENA repository.
> 5. We thank the reviewer for this suggestion. We have modified this adverb to prevent confusion.

---

> > ### Author Response · Authors · 2024-11-28
> >
> > We sincerely thank the reviewer once again for their insightful suggestions and comments. As highlighted in the first comment’s update, the revised manuscript incorporates several experiments aimed at the biological validation of the reported results. For example, we have provided a bibliographic validation of the inferred causal graph and included an experiment to assess its reliability and robustness. Specifically, we demonstrate that the majority of edges in the learned graph maintain their sign, thereby preserving the direction of causality. This finding supports the plausibility of the inferred causal mechanisms. A detailed analysis of this study is available in Appendix II.
> >
> > Furthermore, we have evaluated the biological patterns captured at the latent factor level using Level 2 GO Biological Processes (BPs). This analysis reveals clusters of latent factors that encode true high-level biological processes, potentially addressing the reviewer’s concern regarding biological validation. Additionally, we have expanded the evaluation by including a second dataset, Wessels2023, which we believe helps address the concern of restricted evaluation.
> >
> > We hope these additions address the reviewer’s concerns comprehensively and would be happy to clarify or answer any questions that may arise in the coming days.

---

### Official Review · Reviewer_YiHQ · 2024-11-03

**Soundness:** 3
**Presentation:** 3
**Contribution:** 2
**Rating:** 6
**Confidence:** 4

**Summary:**

This work proposes to extend the discrepancyVAE interventional causal representation learning framework to biological processes applications. Specifically, the authors propose to embed prior knowledge about biological processes (BPs) through a framework called SENA-discrepancyVAE, which recovers latent factors that are a linear combination of a set of biological processes (pathways). The main idea presented in this work is to design a more flexible encoder class (SENA-\delta) specific to mapping biological pathways to latent causal factors for interpretability. Empirical results show that the framework is shown to improve performance in predicting the effect of unseen perturbations.

**Strengths:**

- The paper is well written with strong motivations behind using CRL techniques for biological applications.
- The metrics proposed (differential activation, Hits@N) seem to be robust indicators of perturbation effects on BPs and downstream effects. I believe these evaluation metrics are one of the key interesting contributions of this work.
- The empirical evaluation is exhaustive and illustrates some interesting observations, especially the representational capacity of the VAE-based SENA method compared to the traditional discrepancyVAE.
- The interpretability analysis of the reparameterization layer is interesting and reveals which genes were affected the most upon perturbations. I do believe that exploring real-world applications of CRL is a very important direction.

**Weaknesses:**

- Although the application in gene regulatory networks is quite interesting, this work seems to be more of an evaluation study of the discrepancy-VAE framework proposed by Zhang et al. I do not see much of an added contribution beyond the original paper besides highlighting the application.
- The difference in performance between the SENA variant and the original discrepancyVAE seems to be quite marginal in terms of representation in the double-perturbation scenario. For instance, in Table 2, the KL-divergence for double-perturbation prediction is only marginally better than the original MLP-based discrepancyVAE.

**Questions:**

What is the intuition behind the $\lambda$ hyperparameter to tune small influences of a gene on a biological process? Should this be a constant value throughout the mask matrix or would it be better to learn this influence via some type of attention weights?

---

> ### Author Response · Authors · 2024-11-22
>
> **Weaknesses:**
> 1. We understand the concerns raised by the reviewer. In the new version of the manuscript we have clearly stated the significant value brought by the proposed model, and its differences with the discrepancy-VAE. In brief,  we have demonstrated how biological processes can be used as prior knowledge in the context of causal representation learning. The resulting model, SENA-discrepancy-VAE, is on par, or even outperforming it in specific scenarios, in terms of predictive capabilities with the original discrepancy-VAE, while at the same time producing embeddings that can be easily inspected for assessing their biological meaning. Thus, in our opinion, and as mentioned by other reviewers, the proposed model nicely closes the gap between identifiability and interpretability which was long needed in the field of causal representation learning in biomedicine.
>
> 2. We would like to note that the main goal of the proposed model was not to outperform the original model in terms of reconstruction capabilities but to provide the long-needed capabilities to fully interpret the causal latent factor in the context of biomedical research, while providing similar reconstruction capabilities and identifiable guarantees as in the discrepancy-VAE.  Interestingly, and despite the restrictions imposed by the SENA-δ encoder that could potentially decrease the SENA-discrepancy-VAE representational capabilities, the proposed model outperformed the MLP encoder for some latent dimensions in terms of MSE and MMD computed on unseen double perturbations for small values of λ (0.1). Moreover, setting λ = 0 allowed the SENA-discrepancy-VAE to surpass the original MLP encoder on the DKL metric, while the optimal model for causal graph sparsity (L1) varied with latent dimensions. We believe that these results, which align with those of the ablation studies, highlight the potential of the proposed SENA-discrepancy-VAE.
>
> **Questions:**
>
> 1. The intuition behind the lambda parameter is that residual connections between genes and gene sets can better respond to the underlying biological structure of biological processes. Gene sets are a summary of our current knowledge on how biology works. Consequently, some gene sets may be incomplete, with genes involved in the corresponding biological process not included in the gene set. The lambda parameter allows us to overcome this issue, by considering possible contributions from  genes outside the gene set.  Also, we have shown that a small value of lambda can boost the performance without degrading the interpretability. Moreover, we find this experiment really interesting and we are planning to include an analysis where we treat the mask matrix as learning parameters, conditioned to have certain regularization to maintain interpretability. We will present this analysis on the final version of the manuscript.

---

> > ### Comment · Reviewer_YiHQ · 2024-11-28
> >
> > I thank the authors for taking the time to provide clarifications and new results. I do believe that applications of CRL in biology are quite important. The authors have provided more results that have addressed my main concerns. Thus, I raise my score to 6.

---

> > > ### Author Response · Authors · 2024-11-28
> > >
> > > We appreciate the suggestions and comments made by the reviewer, and we are glad to have addressed their main concerns.

---

### Official Review · Reviewer_Gg4s · 2024-11-04

**Soundness:** 3
**Presentation:** 4
**Contribution:** 3
**Rating:** 6
**Confidence:** 3

**Summary:**

The paper addresses the challenge of learning interpretable causal representations for Perturb-seq data (gene expression in cells). The primary contribution is the novel introduction of masking to incorporate biological process (BP) knowledge into an existing method for causal representation learning (discrepency-VAE), which is named SENA-discrepancy-VAE. The masking ensures that latent factors can be interpreted as linear combinations of the activity of BPs. Since this modification is compatible with the discrepancy-VAE, the original model's theoretical guarantees for causal representation learning remain.

The method and ablated variants are evaluated on a Perturb-seq dataset collected from one particular cell line and is set up to minimize the overlap between the BPs. The results demonstrate that SENA performs similarly to discrepency-VAE in terms of reconstruction yet results in sparser and more interpretable results. Furthermore, by studying the contrast between inferred activity levels on perturbed and control samples the authors show that the latent factors can be associated with BPs and are therefore interpretable.

**Strengths:**

- Clear technical contribution that bridges causal representation learning with biological interpretability while maintaining theoretical guarantees
- The paper contributes to causal representation learning for Perturb-seq data by introducing biological interpretability through pathway information, while maintaining the theoretical guarantees of discrepency-VAE.
- Well written and clear presentation of the method and results.
- Thorough ablation studies
- Demonstrates interpretability of latent factors with concrete biological examples.

**Weaknesses:**

- Experimental validation is limited to one dataset and no baselines other than their own ablations and discrepancy-VAE. The paper would benefit from comparisons to at least one of the other listed related works.
- No comparison with simpler approaches like post-hoc interpretation of standard discrepancy-VAE latent factors.
- While the link between latent factors and BPs is investigated, the quality of the discovered causal graph is not.
- Given that the latent factors group a large number of BPs into a small number of latent factors there should be a deeper investigation of the biological plausibility and practicality of this result beyond the contrasting activations.
- Readability of several figure texts should be improved.

**Questions:**

- How sensitive is the model to the quality and completeness of the pathway knowledge used? Have you tested with different pathway databases or subsets of pathways?
- How does the computational complexity scale with the number of biological processes? Is there a practical limit to how many processes can be incorporated?
- Have you explored whether the causal relationships discovered by the model align with known biological pathway interactions beyond the examples provided?
- Can you confirm that all genes involved in the double perturbations were also present in your single-perturbation training data?
- Has $N$ and $\tau$ been mixed up in the Hits@N metric?
- How does table 2 show that “both models tend to assign most interventions to a small number of latent factors”?

---

> ### Author Response · Authors · 2024-11-22
>
> **Weaknesses:**
>
> 1. We thank the reviewer for highlighting this limitation. Regarding the concern about datasets, we have already incorporated a second one, Wessels2024, into the paper and we are working to incorporate a third one if time allows it. On the other hand, we are working on incorporating a state-of-the-art model for perturb seq expression prediction, GEARS, that will allow us to better evaluate the proposed model in terms of prediction of transcriptomic effect of unseen perturbations.
>
> 2. Note that the standard discrepancy-VAE model does not allow interpretability on a geneset-granularity level, however, we are working on including an experiment that evaluates whether the learnt latent factors by the standard discrepancy-VAE are interpretable in the context of the perturbed genes.
>
> 3. Upon inspecting the causal graph, a first important connection is the one between factor 15, “hydrogen peroxide biosynthetic process” which causes factor 69, “endothelial cell morphogenesis”. It is well known that hydrogen peroxide stimulates endothelial cell proliferation [https://pubmed.ncbi.nlm.nih.gov/12572854/][https://www.nature.com/articles/s41598-018-36769-3]. Thus, our causal graph captured this regulatory relationship in a fully unsupervised, data-driven way.
> In turn, factor 53 causally influences factor 15, and factor 53 contains the biological process “catechol-containing compound biosynthetic process”. It is well known that H2O2 can be produced by the metabolism of catecholamines [https://doi.org/10.1016/0006-8993(94)91525-3][https://pubmed.ncbi.nlm.nih.gov/7108528/].
> An even more direct connection exists between latent factor 69 and later factor 2, with the latter including “negative regulation of endothelial cell apoptotic process” among its biological processes.
> Taken together, these findings provide evidences for the correctness of our approach and its capability of recapitulating known biological causal relationships
>
> 4. We thank the reviewer for the interesting question, and we acknowledge the limited analysis on this topic. We are incorporating this experiment treating the biological plausibility as how the latent factors maintain the biological structure of the GO terms in higher aggregation levels. It will be included on the final version of the manuscript.
>
> 5. We have made sure the text within the figures are now more readable and understandable.
>
> **Questions:**
>
> 1. We have empirically validated that the reconstruction loss is similar to the one we show in the manuscript when the number of pathways included vary. In fact, as we showed in the ablation studies, different values of lambda (which means the contribution of genes to gene sets largely increase) yield similar values of reconstruction, hence the main effect may be in the interpretability of the established pathways.
>
> 2. We thank the reviewer for this question. Since biological processes are incorporated to compress gene information in a biologically-driven manner, their number will hardly be a limiting factor (e.g., the total number of BPs in GO is 24K). Nevertheless, we plan to include an analysis on performance and time evaluation when varying the number of biological processes included.
>
> 3. Thanks for the comment. Indeed, evaluating (causal) relationships between biological processes, besides  the (non causal) hierarchical structure defined in GO, is complex and usually necessitates in vitro experiments. While we believe that in vitro experiments are out of the scope of this work,  we are planning to go through the pathway list and look for potential “cascade” effects on the uncovered causal mechanisms.
>
> 4. Yes, we confirm every gene involved in the double perturbations is also present as a single perturbation in the training data.
>
> 5. Thanks for pointing this out. Indeed, we have corrected the formula to use N as the number of positions the BPs are ranked against, getting rid of tau.
>
> 6. Thanks for pointing out this typo. Figure 8 is the one that shows the mappings between perturbations and latent factors, according to the categorical encoding, for SENA and the standard discrepancy VAE. We have corrected this in the text.

---

> > ### Author Response · Authors · 2024-11-28
> >
> > We sincerely thank the reviewer once again for their valuable suggestions and comments. We believe the updated version of the manuscript addresses most of the concerns raised, including:
> >
> > 1. The analysis of latent factors as biologically-driven aggregations of BPs.
> > 2. The robustness and bibliographic interpretability of the discovered causal graphs.
> > 3. The improved readability of certain figures.
> > 4. The inclusion of additional datasets to evaluate the proposed approach.
> > 5. The time complexity analysis as a function of the incorporated BPs.
> >
> > These points have been addressed in detail in the update to the first comment. We would be happy to answer any further questions regarding this revised version of the manuscript.

---

### Author Response · Authors · 2024-11-22
**Global Response to Reviewers - Part 1**

We would like to thank the reviewers for their comments and suggestions, we believe they have precisely pinpoint the strengths and things to improve in our work. For instance, several reviewers commended the novel technical contribution that bridges causal representation learning (CRL) with biological interpretability, leveraging pathway knowledge while maintaining theoretical guarantees, hence **“addressing a pressing need in biomedicine for interpretable CRL”**. Reviewers also praised the **“thoroughness of ablation studies”**, and the proposed robust evaluation metrics, which were seen as **“one of the key interesting contributions of this work”**. Additionally, the visualization of pathway-specific activity levels and causal graphs **“enhances transparency, allowing readers to trace latent factors back to biological functions”**. Overall, we believe reviewers agreed that the proposed model has significant practical impact and since **“interpretability is crucial”**, and **“it makes the model more useful for domain experts”**.

On the other hand, reviewers raised some weaknesses in the work, such that the **“experimental validation is limited to one dataset”**, raising concerns about its generalizability across different cell types or biological contexts. Reviewers also suggested comparisons with **“simpler baselines and other related works”**, which would strengthen the contribution claims. Additionally, we agree with the reviewers that the impact of the λ parameter on model performance, necessitate further **“guidance and sensitivity analyses”**, as it would address the current **“static pathway relevance across all tasks”**, which potentially oversimplifies complex biological interactions. Another mentioned limitation was the absence of **“deeper investigation of the biological plausibility”** of how the grouping of pathways into latent factors aligns with known biological mechanisms. Finally, reviewers suggested improving readability in figures and clarifying specific details in the manuscript.

---

> ### Author Response · Authors · 2024-11-22
> **Global Response to Reviewers - Part 2**
>
> We believe these concerns can be addressed in the current review process. Due to the time constraint, and to facilitate the discussion with the reviewers, we now present a first updated manuscript with several improvements and contributions proposed by the reviewers (see below). We plan to work on the remaining questions and concerns in the following days.
>
> 1. Figure 1 (model overview) has been updated to better reflect the contribution of SENA to the standard discrepancy VAE, clarifying the different modules within this model.
> 2. A second dataset has been added, Wessel2023, which we describe in depth (and compare against Norman2019 dataset, Figure 5) in Supplementary Note II and present benchmarking results in Supplementary Table 3. We are working on refining Figure 5 subfigures and include 5 seeds to the presented results in Table 3.
> 3. A novel analysis has been included (Fig 4) on the Norman et al dataset that further validates  SENA’s capacity to naturally learn biologically-driven patterns without specifically enforcing them.
> 4. Causal graph has been further investigated and bibliographically-validated.
> 5. Minor typos on mathematical notation (e.g. Hits@K) and text (e.g. faithfulness) has been addressed. Also, some citing typos (Figure 8 instead of Table 2, conclusion) has been addressed as well. Moreover, values of Table 1 have been correctly updated.
> 6. Readability of several figure texts has been improved.
> 7. Mathematical notion of Supp Note 1 has been cleaned and transformed into matrix form for the sake of clarity.
>
> As mentioned above, we are planning to upload the final version of the manuscript incorporating remaining concerns raised by the reviewers, which require extra working hours:
>
> 1. We are incorporating a third dataset, Replogle2020 (Nature Biotechnology, Replogle et al 2020), which contains a different type of knock out technology (Single-cell CRISPR) and can provide further generalization capabilities to the manuscript.
> 2. We are including another state-of-the-art model on perturbations prediction, named GEARS (Nature Biotechnology, Roohani et al 2023), for a more robust benchmarking.
> 3. In order to evaluate the biological plausibility of the GO terms into high-level order aggregations, we are providing an analysis on the defined DA score over the latent factors of SENA to evaluate if the mapping between genesets and latent factors can aggregate GOs in a biologically-meaningful manner.
> 4. We are including an analysis on performance and time evaluation when varying the number of biological processes included.
> 5. In order to provide an extra layer of flexibility to the developed SENA’s model, we are including an analysis where we treat the introduced λ as a learning parameter, conditioned to have certain regularization to maintain interpretability.
> 6. We are planning to extend the ablation studies to further find an optimal range of λ across datasets, and propose that as the default value. Moreover, we would pinpoint this in the main manuscript.
> 7. We are extending the ablation studies to evaluate D_KL as a function of λ, and incorporating the randomization of pathway annotation experiments into them.
> 8. We are incorporating a small ablation study on the causal graph generation to evaluate sensitivity of edge weights as a function of λ and latent dimension.
> 9. We will include an experimental validation of the obtained expectation in Equation (10) , providing robust theoretical results to the proposed approach.
> 10. In order to compare the proposed interpretability analysis with the standard discrepancy-VAE, we are performing a post-hoc interpretability on the latent factors of standard discrepancy-VAE to provide benchmarking at the latent level.

---

> > ### Author Response · Authors · 2024-11-28
> > **Global Response to Reviewers - Final Updates**
> >
> > We sincerely thank the reviewers once again for their insightful comments and valuable suggestions throughout the review process. As noted, we have incorporated additional experiments to address the concerns and limitations highlighted by the reviewers. Below, we present an updated and final list of the experiments performed:
> >
> > 1. We have incorporated Wessels2023, a CRISPR-Cas13 perturbation dataset we describe in depth in Appendix V, and compare against the Norman2019 dataset. Results of the benchmarking across SENA and standard discrepancy-VAE shows that higher values of λ are required to obtain similar MMD and MSE scores to the original approach. Also, the analysis on transcriptomic profiles over Wessels2023 suggest that multiple double perturbations are heavily skewed towards a single-perturbation effect, which can justify the difficulty of uncovering interpretable results. See Appendix V for a detailed analysis.
> >
> > 2. A novel analysis on the Norman et al. dataset has been included that further validates SENA’s capacity to naturally learn biologically-driven patterns without specifically enforcing them  (Fig. 4).
> >
> > 3. The inferred causal graph has been further investigated and bibliographically-validated. Moreover, we have incorporated a detailed study on the robustness of the inferred causal graph. We found  that most edges are robust in terms of sign consistency across several latent dimensions and λ values, underscoring the reliability of the inferred causal graph. See Appendix II for further details.
> >
> > 4. Minor typos on mathematical notation (e.g. Hits@K) and text (e.g. faithfulness) has been addressed. Also, some citing typos (Figure 14 is now correctly cited in conclusion) has been addressed as well. Moreover, values of Table 1 have been updated increasing the number of seeds from 3 to 5. Mathematical notion of Appendix I has been cleaned and transformed into matrix form for the sake of clarity. Several figure’s  (Fig. 1, Fig. 2) readability has also been improved.
> >
> > 5. We have included GEARS, a state-of-the-art model for predicting multigene perturbations at the transcriptomic level in the benchmarking. Even though this method does not provide a causal graph, we have computed and reported its MMD performance in Table 1. Overall, GEARS seems to fail to predict unseen double-perturbations for the evaluated Norman2019 dataset. We have further described this evaluation in Appendix VI.
> >
> > 6. We have included an analysis on the aggregation of SENA’s Biological Processes (BP) into high-level scores by measuring the contribution of specific groups of BP, according to the level 2 Gene Ontology BPs, to every latent factor, across several latent dimensions and λ values. We showed that multiple meta pathways (that is, latent Zs) were significantly associated with specific level 2 pathways, underscoring our model capabilities to learn biologically-meaningful patterns at both high (BPs) and broad (meta-pathway) granularities. See Appendix III for further details.
> >
> > 7. We have incorporated an analysis on time complexity and KLD performance across different groups (from 1 to ~1000) of BPs, by enforcing a minimum number of genes within each BP. Results are shown in Figure 10 (Appendix IV).
> >
> > 8. We have provided an experimental study on the derived Eq.10-11, showing that it holds across perturbations, λ parameters and even models (SENA-discrepancy-VAE vs standard discrepancy-VAE). See Appendix I and Figure 5.

---

### Meta-Review · Area_Chair_ygWh · 2024-12-22

**Metareview:**

This paper proposes SENA-discrepancy VAE, a causal representation learning model with latent variables linked to biological processes. This is achieved by representing each latent factor as the linear combination of biological processes. The proposed method is shown effective on a biological dataset. The reviewers have concerns on the originality of this work compared to discrepancy VAE and the limited experimental evaluation on a single dataset. The authors addressed these concerns and added an additional dataset. All reviewers are positive about this paper after rebuttal and discussion. I agree with the reviewers and recommend acceptance of this paper.

**Additional Comments On Reviewer Discussion:**

The main concerns centered around the limited novelty of this work compared to discrepancy VAE and the limited experimental evaluation on a single dataset. In the rebuttal, the authors convinced the reviewers that the biological extension of the discrepancy VAE model is of practical significance in biology. Also, the experiments on one additional dataset are added to the revised version. The authors also mentioned that they are doing real biological experiments to verify the ideas in this paper. Two reviewer changed their rating from negative to positive after the rebuttal and discussion. I think this paper may have a great impact on CRL in the biological science field.

---

### Decision · Program_Chairs · 2025-01-22

Accept (Poster)